# Dermis resident macrophages orchestrate localized ILC2 eosinophil circuitries to promote non-healing cutaneous leishmaniasis

Sang Hun Lee[1], Byunghyun Kang[2], Olena Kamenyeva [3],
Tiago Rodrigues Ferreira [1], Kyoungin Cho[4], Jaspal S. Khillan[4], Juraj Kabat[3],
Brian L. Kelsall [2] & David L. Sacks[1] ✉

Tissue-resident macrophages are critical for tissue homeostasis and repair. We previously showed that dermis-resident macrophages produce CCL24 which mediates their interaction with IL-4+ eosinophils, required to maintain their M2-like properties in the $T_H1$ environment of the *Leishmania major* infected skin. Here, we show that thymic stromal lymphopoietin (TSLP) and IL-5+ type 2 innate lymphoid cells are also required to maintain dermis-resident macrophages and promote infection. Single cell RNA sequencing reveals the dermis-resident macrophages as the sole source of TSLP and CCL24. Generation of *Ccl24-cre* mice permits specific labeling of dermis-resident macrophages and interstitial macrophages from other organs. Selective ablation of TSLP in dermis-resident macrophages reduces the numbers of IL-5+ type 2 innate lymphoid cells, eosinophils and dermis-resident macrophages, and ameliorates infection. Our findings demonstrate that dermis-resident macrophages are self-maintained as a replicative niche for *L. major* by orchestrating localized type 2 circuitries with type 2 innate lymphoid cells and eosinophils.

Virtually every tissue is populated by one or more distinct populations of tissue-resident macrophages (TRM) which are critical for the maintenance of tissue homeostasis and functions[1,2]. Each of these TRM populations, localized to distinct microanatomical niches, differs in their ontogeny, rate of replacement by monocyte-derived cells, and capacity for self-renewal. TRMs in the brain are embryonically seeded, long-lived, and self-renewing to maintain their homeostatic pool without contribution from cells of hematopoietic origin, whereas others, including those in the intestine, pancreas, lung, heart and dermis, are seeded embryonically but are continuously replenished

after weaning by monocyte-derived cells[3–7]. This mosaic composition can change throughout life by various inflammatory and infectious challenges[8].

The association of embryonic-derived TRMs with M2-like functions has been proposed in various sterile injury and infection models. Embryonic-derived cardiac and arterial TRMs, for example, were found to produce minimal inflammatory responses and instead drove tissue regeneration after sterile injury[9]. In the lung and skin, embryonic-derived macrophages were shown to have M2-like characteristics that better supported the growth of *Mycobacterium tuberculosis* and

[1]Laboratory of Parasitic Diseases, National Institute of Allergy and Infectious Diseases, National Institutes of Health, Bethesda, MD 20892, USA. [2]Laboratory of Molecular Immunology, National Institute of Allergy and Infectious Diseases, National Institutes of Health, Bethesda, MD 20892, USA. [3]Biological Imaging Section, Research Technology Branch, National Institute of Allergy and Infectious Diseases, National Institutes of Health, Bethesda, MD 20892, USA. [4]Mouse Genetics and Gene Modification Section, Comparative Medicine Branch, National Institute of Allergy and Infectious Diseases, National Institutes of Health, Rockville, MD 20852, USA. ✉e-mail: dsacks@niaid.nih.gov

*Leishmania major*, respectively, in comparison to monocyte-derived cells[7,10]. The type 2 cytokines required for sustaining alternatively activated TRMs have been investigated in adipose tissue[11,12]. Active phagocytosis of cellular material also imprints TRMs toward an anti-inflammatory phenotype[13,14]. We previously proposed a cooperative interaction between dermis TRMs and eosinophils which functioned to maintain the number and anti-inflammatory program of dermis TRMs in an infection-driven $T_H1$ environment[15]. During *L. major* infection, eosinophils were shown to be the major source of IL-4 required to mediate the local proliferation of dermal TRMs and maintain their M2-like phenotype. The IL4-stimulated dermal TRMs produced a large amount of CCL24 which amplified the recruitment of eosinophils.

Type 2 Innate lymphoid cells (ILC2s), which are strategically located at mucosal barriers, have been identified as major inducers of type 2 responses against allergens or helminth parasites[16]. ILC2s secrete large amounts of IL-5, IL-13, and amphiregulin, which induce essential components of type 2 inflammation, such as goblet cell metaplasia, mucus production, eosinophils/mast cell activation, and alternative activation of macrophages. A growing body of evidence indicates that resident ILC2s in lung and adipose tissue promote M2-like TRMs during metabolic homeostasis as well as $T_H2$-associated infection[17,18]. In the skin, ILC2s were the first ILCs to be discovered both in mice and humans[19,20]. ILC2-derived cytokines in the skin play an essential role in atopic dermatitis, scleroderma and other fibrotic skin disorders[21]. The epithelial cytokines IL-33, IL-25 and thymic stromal lymphopoietin (TSLP) are strong ILC2 activating ligands, often characterized as "alarmins" that are released by the barrier epithelium in response to external insults[22]. As such, epidermal alarmins have been shown to mediate critical intercellular interactions between the outermost barrier, epidermal keratinocytes and downstream immune effectors via ILC2s[23]. The role of ILC2s and alarmins in dermal sites of infection, including infection with vector borne pathogens, has not been investigated.

In this study, we utilize single cell RNA sequencing (scRNA-seq) to characterize dermal TRMs in the *L. major* infected skin, and identify an unappreciated type 2 circuitry involving eosinophils and ILC2s, and orchestrated by MHCII⁻MR^high dermal TRMs. Disease progression is critically dependent on IL-5 production by ILC2, and on TSLP receptor signaling, but independent of either IL-25 or IL-33. By generating *Ccl24-Cre* mice, we confirm that *Ccl24* expression in the lesion is confined to MHCII⁻MR^high dermis TRM, and show that *Ccl24* expression selectively mark TRMs in many steady state tissues. By generating *Tslp^f/f* mice and crossing with *Ccl24-Cre* mice, we conditionally ablate TSLP expression by MHCII⁻MR^high dermis TRMs to show that they are the major source of TSLP required for the evolution of non-healing cutaneous disease.

## Results

### IL-5 from ILC2s is required for dermal TRMs and non-healing infection

We previously reported that during infection IL-5 is produced mainly by ILC2s and is critical to mediate the non-healing response to *L. major* Seidman strain (LmSd)[15]. Here, we confirm that ILC2s are the main producers of IL-5 by analyzing tdTomato expression in *R5: ROSA26-LSL-tdTomato* mice, which fate-maped cells expressing the *Il5* promoter-linked Cre recombinase[24] (Fig. 1a). We failed to detect

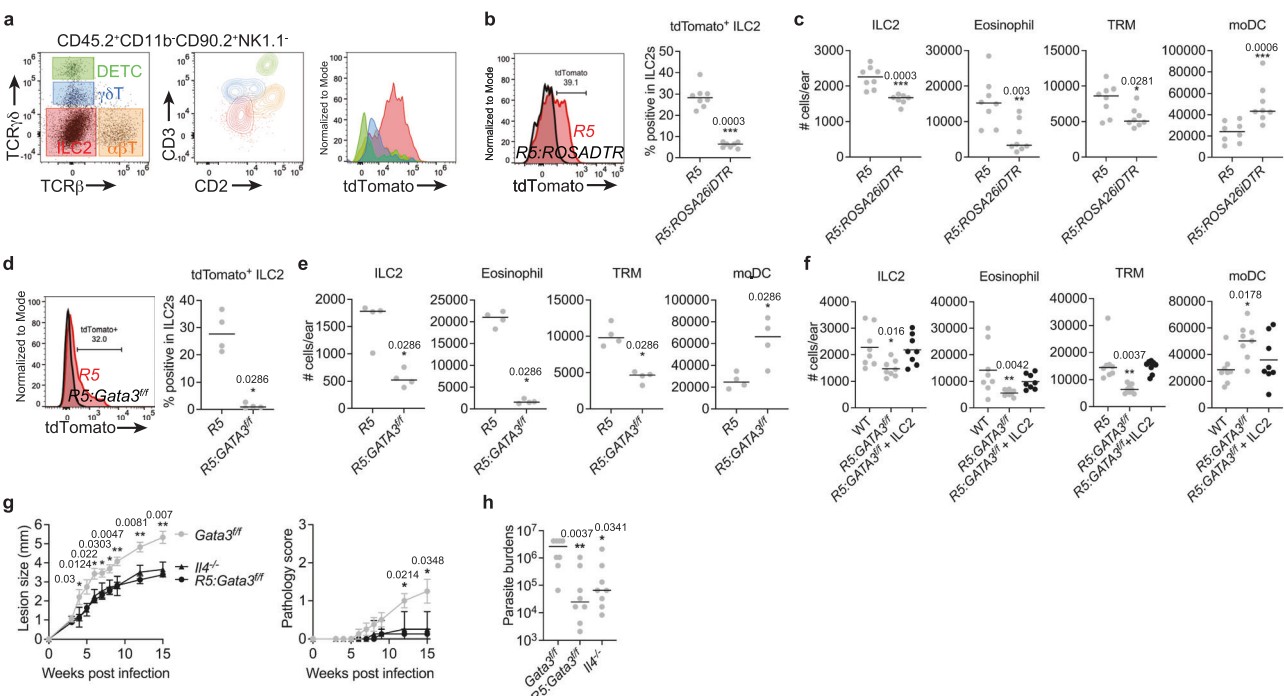

**Fig. 1 | IL5⁺ ILC2s are required to maintain M2-like dermal TRMs which play a critical role in non-healing *L. major* infection. a** Representative flow cytometric analysis of ear isolates prepared from naive *R5: ROSA26-LSL-tdTomato* mice. CD45.2⁺CD11b⁻CD90.2⁺NK1.1⁻-gated cells were analyzed for the expression of TCRβ and TCRγδ, and subdivided into DETCs (green; TCRγδ^high TCRβ⁻), γδT cells (blue; TCRγδ^int TCRβ⁻), ILC2s (Red; TCRγδ⁻TCRβ⁻), and αβT cells (orange; TCRγδ⁻TCRβ⁺). Both CD2/CD3 and tdTomato expression were shown by overlaying color-coded populations in CD2-CD3 axes and histogram respectively. **b** tdTomato expression in ILC2s and (**c**) the absolute numbers of indicated cells recovered from ears of *R5* (n = 8) vs. *R5: ROSA26iDTR* (n = 7) mice were measured at 12 days p.i. with 2 × 10⁵ LmSd followed by DT treatment. **d, e** The same parameters are shown as above in ear isolates prepared from *R5* (n = 4) vs. *R5: Gata3^f/f* (n = 4) mice at day 12 p.i. with 2 × 10⁵ LmSd, or (**f**) WT (n = 8) vs. *RT: Gata3^f/f* (n = 8) vs. *R5: Gata3^f/f* (n = 8) mice adoptively transferred with in 0.5 × 10⁶ in vitro expanded, splenic ILC2s from WT mice, intravenously administered on day 0 p.i. with 2 × 10⁵ LmSd. **g** Lesion development and pathology score over the course of infection with 10³ LmSd metacyclic promastigotes in the ear dermis of *Gata3^f/f*, *Il4⁻/⁻*, and *R5: Gata3^f/f* animals (n = 8). **h** Parasite burdens were quantified at 12 weeks p.i. Values represent mean ± standard deviation (n = 8). *P < 0.05, **P < 0.01, and ***P < 0.01 by two-sided non-parametric Mann-Whitney test (**b–e**) and one-way ANOVA with Dunn's posttest compared with *Gata3^f/f* (**f, g**). Data are representative of two independent experiments (**a–g**). Source data are provided as a Source Data file.

tdTomato expression in other lymphoid cells. To ablate IL-5 producing ILC2s, we administered diphtheria toxin (DT) to *R5: ROSA26iDTR* mice, which resulted in the disappearance of tdTomato⁺ ILC2s from ear skin (Fig. 1b). Since IL-5 contributes to the maturation and recruitment of eosinophils, and since eosinophils are a critical source of IL-4 to maintain dermal TRMs through local proliferation[15], we determined the number of eosinophils and dermal TRMs in the IL5⁺ ILC2-depleted animals (Fig. 1c). Each of these populations was significantly reduced in the *R5: ROSA26iDTR* mice, while inflammatory monocyte-derived cells (moDCs) were increased in the lesions at 12 days post-infection (p.i.) compared to *R5* mice. As repeated injection of DT to deplete IL-5⁺ ILC2s for an extended period induces an anti-DT response[25], we generated *R5: Gata3^{f/f}* mice which lack an essential ILC2 transcription factor in IL5⁺ ILC2[26]. We observed the complete loss of tdTomato⁺ ILC2 (Fig. 1d) and significant decrease in total ILC2s, eosinophils, and dermal TRMs (Fig. 1e) in the *R5: Gata3^{f/f}* mice compared to *R5* mice. The diminished numbers of TRMs and eosinophils in the infected, *R5: Gata3^{f/f}* mice was effectively restored by adoptive transfer of expanded ILC2s from WT mice, while the elevated number of moDCs were reduced to levels comparable to those observed in WT controls (Fig. 1f). When challenged with a low dose of LmSd, *Gata3^{f/f}* control mice developed non-healing, ulcerative lesions, while lesion progression and pathology were substantially ameliorated in the *R5: Gata3^{f/f}* mice, similar to the *Il4⁻/⁻* mice (Fig. 1g). Additionally, there was a 60-fold reduction in lesion parasite burden in *R5: Gata3^{f/f}* compared to control mice (Fig. 1h).

## TSLPR signaling is required for ILC2s and non-healing infection

To define the role of three epithelial alarmins in promoting non-healing cutaneous infection, we challenged WT, *Il25⁻/⁻*, *Il33⁻/⁻*, and *Tslpr⁻/⁻* mice with a low-dose of LmSd. Genetic deletion of TSLPR significantly ameliorated disease progression as measured by lesion size, pathology, and parasite burden (Fig. 2a, b). By contrast, neither *Il25⁻/⁻* nor *Il33⁻/⁻* mice showed differences in these parameters compared to WT mice. *Tslpr⁻/⁻* mice showed significantly fewer numbers of ILC2s, eosinophils, and dermal TRMs (Fig. 2c), as well as reduced expression of IL-5 and IL-13 by ILC2s (Fig. 2d).

## Dermal TRMs as the source of TSLP during *L. major* infection

To identify the source of TSLP during cutaneous leishmaniasis, we sorted CD45⁺ hematopoietic and CD45⁻ nonhematopoietic cells from the ears of either naïve or infected WT and *eoCre: Il4/13^{f/f}* mice having a selective deficiency of IL-4/IL-13 in eosinophils. The sorted cells were pooled in a 9:1 ratio (CD45⁺:CD45⁻), and scRNA-seq was performed using the 10x Genomics platform on a total of 17,355 cells (4592 cells naïve WT; 3728 infected WT; 5913 naïve *eoCre: Il4/13^{f/f}*; and 3552 infected *eoCre: Il4/13^{f/f}*). Uniform manifold approximation and projection (UMAP) dimensional reduction analysis with unbiased clustering based on a merged dataset identified 20 distinct clusters from all four groups (Fig. 3a and Supplementary Fig. 1) with differential expression of marker genes (Supplementary Fig. 2A). For identification of each cluster, we used reference-based cell annotation (SingleR), using ImmGen repository as a ref. [27]. Further fine-tuning by examining individual cluster markers identified 17 distinct hematopoietic and 3 nonhematopoietic cell clusters. The cellular annotations were well supported by several lineage specific cell surface protein markers using Cellular Indexing of Transcriptomes and Epitopes by Sequencing (CITE-Seq) (Supplementary Fig. 2B). Proliferating cells defined by cell cycle-specific genes were mainly CD4⁺ T cells expressing CD3 and CD4 CITE-Seq markers. To refine the identities of myeloid clusters which showed broad expression of CITE-Seq myeloid markers such as CD11b, CD11c, F4/80 (Supplementary Fig. 2B), as well as similar transcriptional profiles (Supplementary Fig. 2A; red box), we directly compared our scRNA-seq myeloid clusters to published microarray data from flow cytometry-sorted skin myeloid cells[5]. We repeated principal component analysis (PCA) comparing monocyte lineages and MHCII⁺/MHCII⁻ dermal macrophages from the aforementioned microarray study (Supplementary Fig. 3A). By comparing the top 100 DEGs that contributed to PC1 from this prior study to our scRNA-seq data, we could confirm transcriptional programs that clearly distinguish monocytes and dermal TRMs in the scRNA-seq analysis (Supplementary Fig. 3B). In addition, connectivity MAP (cMAP) scores, which show the degree of relatedness of cell subsets to the selected DEGs[5,28], demonstrated the transcriptional relatedness of the monocyte and dermal TRM clusters in the scRNA-seq analysis to the respective dermal monocytes and

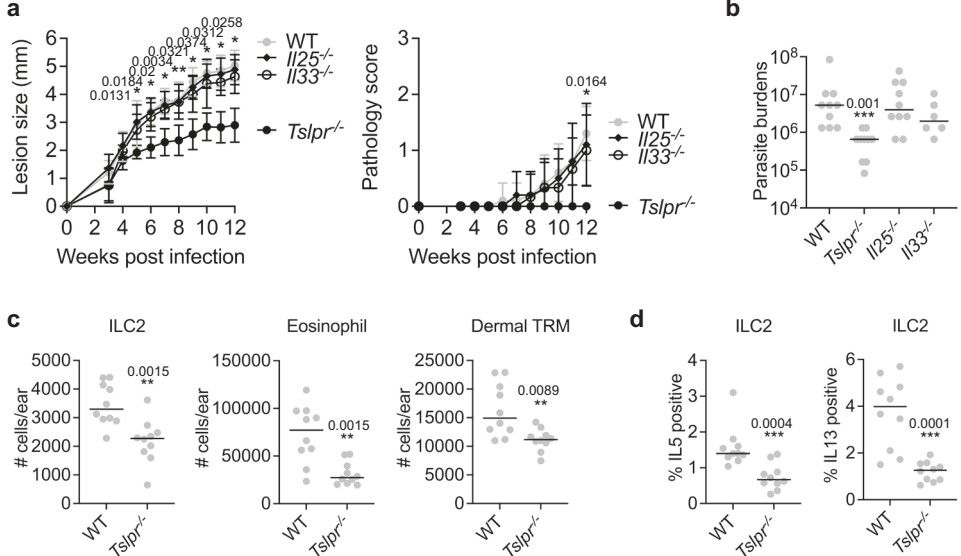

**Fig. 2 | TSLP receptor signaling promotes non-healing *L. major* infection.**
**a** Lesion development and pathology score over the course of infection with 10³ LmSd metacyclic promastigotes, and (**b**) Parasite burdens were quantified at 12 weeks p.i. in the ear dermis of WT (*n* = 10), *Il25⁻/⁻* (*n* = 10), *Il33⁻/⁻* (*n* = 6), and *Tslpr⁻/⁻* (*n* = 10) mice. **c** The absolute numbers of indicated cells and (**d**) intracellular staining for IL5/IL13 expressions in ILC2s from ear isolates of WT and *Tslpr⁻/⁻* mice were compared at 12 days p.i. with 2 × 10⁵ LmSd (*n* = 10). Values represent mean ± standard deviation. *\*P < 0.05, \*\*P < 0.01, and \*\*\*P < 0.001* by one-way ANOVA with Dunn's posttest compared with *Tslpr⁻/⁻* (**a**) and two-sided non-parametric Mann-Whitney test (**b**–**d**). Data are representative of two independent experiments (**a**–**d**). Source data are provided as a Source Data file.

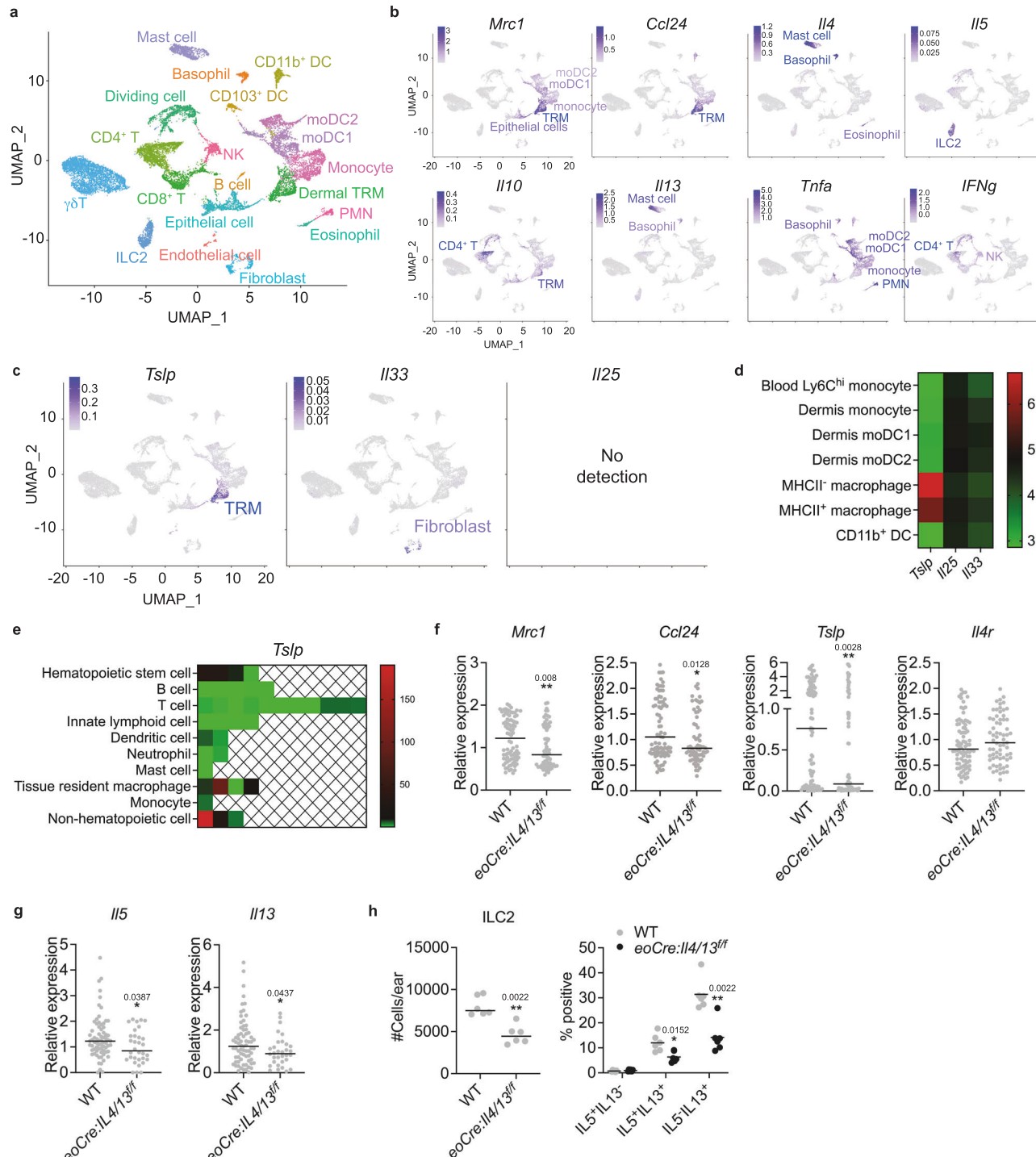

**Fig. 3 | TSLP is expressed by dermal TRMs during *L. major* infection. a** UMAP plot representing of 17,355 cells combined from four samples; naïve WT (4592 cells) and *eoCre: Il4/13^f/f* (5913 cells), and infected WT (3728 cells) and *eoCre: Il4/13^f/f* (3552 cells) mice, 12 days post-challenge with $2 \times 10^5$ LmSd in the ear dermis. **b, c** Selected gene expression UMAP plots. The identities of expressing cells were labeled and color-coded as the magnitude of expressions in scale. **d** Heat map to visualize *Tslp*, *Il25*, and *Il33* expression of indicated dermal myeloid cells, generated from Expression (GEO) database[5]. **e** Comparison of *Tslp* transcription from various immune cells published by Immgen[27]. The relative expression of *Tslp* gene is shown as a heatmap among the indicated populations. Each square represents either a specified subset or population of the specified cell isolated from different organs. **f** Dot plots of expressed transcripts within dermal TRMs from infected WT ($n = 86$) and *eoCre: Il4/13^f/f* ($n = 69$) animals and (**g**) ILC2s from infected WT ($n = 73$) and *eoCre: Il4/13^f/f* ($n = 34$) animals. **h** The absolute numbers and % IL4/IL13^+ ILC2s from ear isolates of WT and *eoCre: Il4/13^f/f* mice at 12 days p.i. with $2 \times 10^5$ LmSd ($n = 6$). Values represent mean ± standard deviation. *$P < 0.05$ and **$P < 0.01$ by two-sided non-parametric Mann-Whitney test (**f**–**h**). Data are representative of two independent experiments (**h**). See also Supplementary Fig. 1, 2, and 3. Source data are provided as a Source Data file.

MHCII$^+$/MHCII$^-$ dermal macrophages in the microarray data (Supplementary Fig. 3C). The moDC1 have a transcriptional signature which is closer to dermal monocytes than to MHCII$^+$/MHCII$^-$ dermal macrophages, while the moDC2 have lost relatedness to dermal monocytes, consistent with our prior observation that adoptively transferred monocytes sequentially differentiate to moDC1 and then to moDC2 during infection[7]. Thus, we identified four myeloid clusters as inflammatory monocytes, two monocyte-derived cells (moDC1 and 2), and dermal TRMs.

The strong expression of both *Mrc1* and *Ccl24* reinforced the identity of dermal TRMs as previously reported[15], while monocytes, their derivatives, and some epithelial cells also showed *Mrc1* expression, though to a lesser degree (Fig. 3b). Mast cells, basophils, and eosinophils expressed *Il4*, while only ILC2s actively transcribed *Il5*. Lastly, *Il10* expression was detected in both CD4$^+$ T cells and dermal TRMs. Of the three epithelial alarmins, only *Tslp* was strongly expressed, and unexpectedly confined to the dermal TRMs. Fibroblasts expressed *Il33*, though at a very low level, and no active transcription of *Il25* was detected in any cell cluster (Fig. 3c and Supplementary Fig. 1C). Revisiting the publicly available microarray data obtained from skin myeloid cells[5], *Tslp* expression was restricted to both MHCII$^+$ and MHCII$^-$ dermal macrophages but was higher in the MHCII$^-$ subset (Fig. 3d). Based on the expression profiles of various hematopoietic and non-hematopoietic populations compiled in the Immgen data repository[27], *Tslp* expression has been detected in other TRMs, including splenic red pulp, peritoneal, and brain macrophages, in addition to hematopoietic stem cells and epithelial cells (Fig. 3e). Interestingly, among the receptors for the three epithelial alarmins, the dermal TRMs expressed only the receptor for TSLP (*Crlf2*) (Supplementary Fig. 3D).

The expression of the key genes involved in the localized type 2 circuitry were plotted to track changes between WT and *eoCre: Il4/13$^{f/f}$* mice having a selective deficiency of IL-4 (and IL-13) in eosinophils. The TRMs from the *eoCre: Il4/13$^{f/f}$* mice displayed significantly lower expression of *Mrc1*, *Ccl24*, and *Tslp* than WT mice after infection, whereas *Il4r* expression remained unchanged (Fig. 3f). ILC2 from *eoCre: Il4/13$^{f/f}$* mice also showed a reduction of *Il5* and *Il13* expression compared to WT mice (Fig. 3g). We confirmed the decreased production at the protein level of IL-5 and IL-13 by ILC2s in *eoCre: Il4/13$^{f/f}$* mice, as well as the reduced number of ILC2s compared to WT mice after dermal challenge (Fig. 3h).

## Tslp and Ccl24 are co-expressed in the MHCII$^-$ subset of dermal TRMs

We performed unbiased sub-clustering within the dermal TRM cluster (Supplementary Fig. 4A). A minor subcluster was defined (cluster 1) that expressed several monocytic markers, including *Ly6c2*, *Ccr2*, *Plac8*, *Cxcl10* (Supplementary Fig. 4B, C). By contrast, the major TRM cluster (cluster 0), expressed higher levels of *Pf1*, *Cd163*, *F13a1*, *Igf1*, *Cd36*, *Csf1r* and *Lyve1*, well-defined markers of TRMs. Evidence that cells in cluster 1 are infiltrating monocytes was further supported by our observation that while the number of cells in cluster 0 was diluted out of the UMAP by infiltrating cells after infection, cluster 1 showed an increase in size upon infection (Supplementary Fig. 4A; bottom panel). Thus, we used only cluster 0 for the downstream analysis of dermal TRMs.

Two ontogenically distinct subsets of dermal macrophages, MHCII$^+$ and MHCII$^-$, have been reported in murine skin[5,29]. To further dissect these populations in our dataset, we generated cMAP-scores for each cell in our dermal TRM cluster to assess their relatedness to the respective MHCII$^+$ and MHCII$^-$ dermal macrophages in the reported microarray dataset[5]. We selected 231 microarray DEGs (105 Up/126 Down, *absolute log2 FC > 1; FDR < 0.05*) from the comparison of MHCII$^+$ and MHCII$^-$ dermal macrophages as a reference gene set. cMAP analysis revealed that 410 (red) and 322 (Gray) cells aligned closely to the transcriptional profiles of MHCII$^-$ and MHCII$^+$ dermal macrophages,

respectively (Fig. 4a). No transcriptional similarity with either subset was exhibited by 137 out of 869 cells. When the populations color-coded by cMAP score were overlaid onto the original UMAP, they were delineated into two distinct clusters within the dermal TRM population (Fig. 4b). Cells within the gray cluster showed high expression of *H2-ab1*, as expected (Fig. 4c). Cells within the red cluster showed strong expression of all three genes of interest, *Mrc1*, *Ccl24*, and *Tslp*. The expression levels of these three genes at the single cell level were strongly correlated in all three paired comparisons, *Mrc1-Ccl24* ($r = 0.835$), *Mrc1-Tslp* ($r = 0.798$), and *Ccl24-Tslp* ($r = 0.827$) (Fig. 4d).

We detected 160 filtered DEGs (*absolute logFC > 0.5* and *adjusted p-value < 0.05*) delineating the MHCII$^-$ and MHCII$^+$ subclusters (Fig. 4e). The MHCII$^-$ cluster showed higher expression levels of genes associated with alternative activation of macrophages (*Maf*), fatty acid oxidation fueling M2 macrophages (*Apoe*, *Cd36*, *Nrp1*, *Glul*), wound healing (*Tgfrb2*) as well as *Lyve1*, identified as a marker for perivascular TRMs across tissues[30]. In contrast, the MHCII$^+$ cluster had increased expression of monocytic *Cd14* and *Ccr2*, pro-inflammatory *Il1b* and TNF pathway related genes (*Nfkb1*, *Traf1*, *Nfkbiz*, *Rel*, *Nfkbia*, *Tnfaip3*, *Tnip3*), and antigen presentation genes (*H2-DMa*, *H2-DMb1*, *H2-Ab1*, *H2-Eb1*, *H2-Aa*, and *Cd74*). Ingenuity pathway analysis (IPA) predicts relatively stronger M2-like characteristics in the MHCII$^-$ subcluster, including anti-inflammatory MSP-RON and PD-1/PD-L1 pathways, lipid metabolic LXR/RXR, and PPARα/RXRα pathways. By contrast, multiple pathways associated with pro-inflammatory responses, including T$_H$1, TLR mediated, and acute phase responses, are associated with the MHCII$^+$ subcluster (Fig. 4f).

## TSLP mediates the interaction between MR$^{high}$ dermal TRMs and ILC2s

We next accessed the location of ILC2 in skin layers by enzymatically separating the epidermis and dermis from intact murine ear skin. Among the skin lymphoid populations, DETCs (dendritic epidermal T cells) reside in the epidermis, whereas most ILC2 and αβ/γδ T cells populate the dermis (Fig. 5a). To visualize ILC2s with dermal TRMs in the steady state dermis in vivo, we performed intravital microscopy (IVM) on the upper dermis of *R5: ROSA26-LSL-tdTomato* mice using Cy5-Manocept$^{TM}$ labeling, previously shown to selectively bind to the mannose receptor on the surface of dermal TRMs[15]. Dermal tdTomato$^+$ ILC2 in the steady state continuously interacted with or were in close proximity to dermal TRMs (Fig. 5b). Some epidermal tdTomato$^+$ ILC2 were spotted in hair follicles extended from the epidermis into the deep dermis (demarcated by dotted lines). IVM imaging also revealed most ILC2 were closely associated with blood vessels in the steady state (Fig. 5b, right panel).

To test the possibility that TSLP controls ILC2 infiltration and interaction with dermal TRMs, we visualized the behavior of ILC2 with or without TSLP neutralization in vivo. After 9 days post-infection, the number of tracked tdTomato$^+$ ILC2 in the field of view was about three-fold fewer in the TSLP-neutralized animal compared with immunoglobulin G (IgG) treated control (25 vs 90 tracked tdTomato$^+$ ILC2s; Fig. 5c left images). Accordingly, the number of contacts between dermal TRMs and tdTomato$^+$ ILC2s was six times fewer in TSLP-neutralized animal vs. control (11 vs 66 tracked contacts; Fig. 5c middle panel). To access the quality of association, we calculated the % surface of tdTomato$^+$ ILC2s which was less than 1 µm distance from the surface of dermal TRMs, which was considered as a physical association. On average, 60% of surfaces of tdTomato$^+$ ILC2s in the control mouse, vs. 40% of their surfaces in the TSLP-neutralized animal, were associated with dermal TRMs during their contacts (Fig. 5c right panel).

## Ccl24-cre mice target MR$^{high}$ dermal TRMs and TRMs from multiple organs
Based on our finding that *Ccl24* is exclusively expressed by dermal TRMs among the cells recovered from the ear of naïve and infected

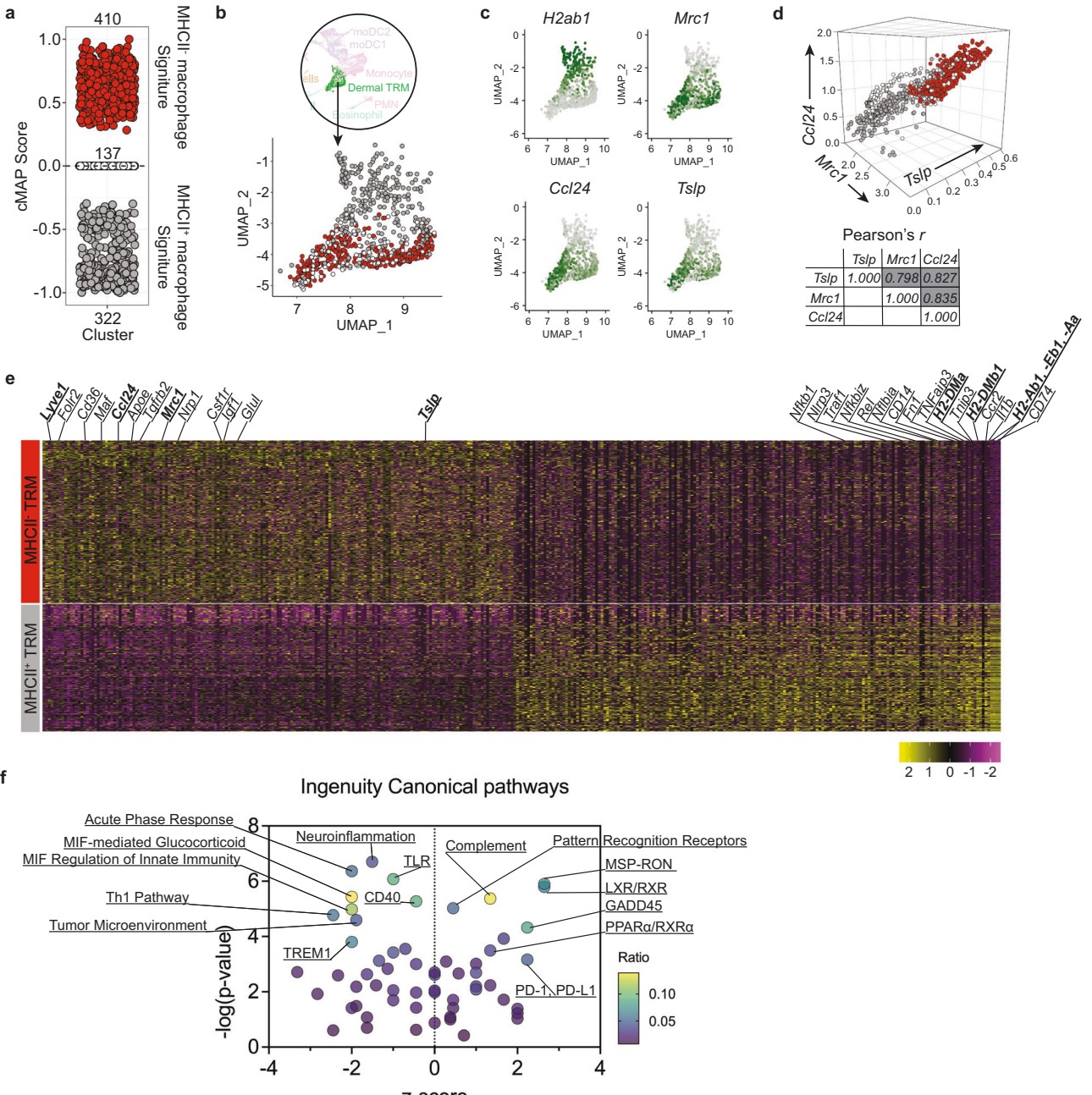

**Fig. 4 | Characterization of MHCII⁺ and MHCII⁻ subpopulation of dermal TRMs in scRNA-seq. a** cMAP analysis of dermal TRMs showing their enrichment for either MHCII⁺ (gray) or MHCII⁻ (red) dermal macrophage transcriptomes (Ly6C^low^CD64^high^MerTK⁺) which were previously published[5]. Cells having transcriptional similarity to neither subset are marked as zero cMAP score (clear). **b** UMAP of dermal TRMs overlayed by MHCII⁺ (gray) and MHCII⁻ (red) subsets color-coded by cMAP analysis in (**a**). **c** Selected gene expression UMAP plots of dermal TRMs. **d** 3-dimensional scatter plot showing statistical correlation between *Mrc1*, *Tslp*, and *Ccl24* gene expressions in MHCII⁺ (gray) or MHCII⁻ (red) dermal TRMs. **e** The heatmap of DEGs (absolute logFC > 0.4 and p_val_adj (FDR corrected < 0.05)). DEGs between two groups of cells were identified using Wilcoxon Rank Sum Test. **f** IPA analysis is performed using the DEGs from the comparison between MHCII⁺ (gray) and MHCII⁻ (red) dermal TRMs. The *p* value of overlap is calculated using the right-tailed Fisher's Exact Test. See also Supplementary Fig. 4.

mice (Fig. 3b), we generated two transgenic animals to confirm dermal TRMs as the critical source of TSLP for the development of non-healing infection: (1) *Ccl24-ires-cre* knock-in mice (*Ccl24-cre*) designed to express poly-cistronic mRNA consisting of *Ccl24* and Cre recombinase genes under the control of the endogenous promoter/enhancer elements of the *Ccl24* locus (Supplementary Fig. 5A), and (2) *Tslp^f/f^* mice having a floxed allele with *loxP* sites flanking exon1 and 2 of the *Tslp* gene, resulting in a frameshift mutation upon cre-mediated deletion (Supplementary Fig. 5B). Of note, integrating the *ires-cre* sequence after the stop codon of the *Ccl24* transcript resulted in 40–45%

reduction of steady-state *Ccl24* expression in this transgenic allele (Supplementary Fig. 5C). Thus, we used heterozygous *Ccl24-cre* animals for all subsequent studies. Importantly, the transcription of *Tslp* in *Tslp^f/f^* animals was comparable to WT animals (Supplementary Fig. 5D). Finally, we confirmed cre-mediated deletion of exon1 and 2 of *Tslp* gene in whole ear DNA isolates from *Ccl24-cre: Tslp^f/f^* mice (Supplementary Fig. 5E)

We bred *Ccl24-cre* mice with *ROSA26-LSL-tdTomato* mice to confirm cre recombinase activity by measuring tdTomato expression in skin isolates at 8 days p.i. with 2 × 10⁵ LmSd-GFP. We found that the

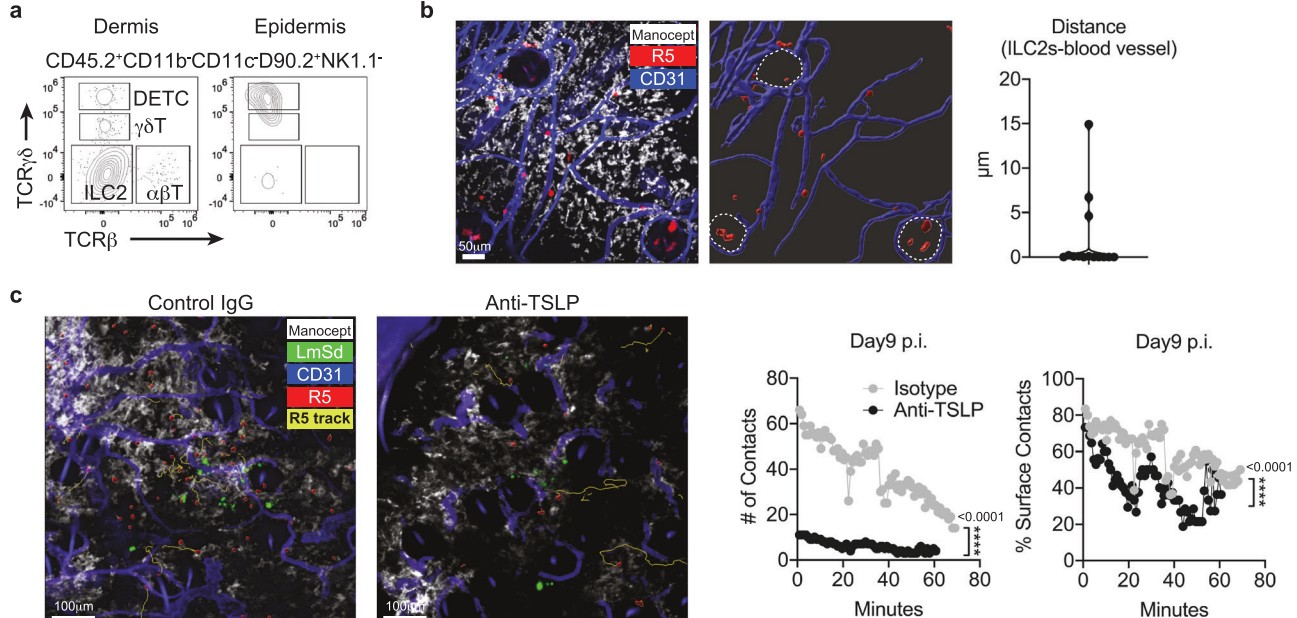

**Fig. 5 | TSLP promotes ILC2 infiltration and interaction with dermal TRMs during *L. major* infection. a** Representative flow cytometric analysis of isolates from epidermal and dermal layers of a naive ear skin to determine the tissue distribution of DETC, γδT, ILC2, and αβT cells. **b** Image obtained from an IVM of the ear of *RS: ROSA26-LSL-tdTomato* reporter mouse intravenously injected with efluor450 anti-CD31 Abs to label blood vessels, and its 3-dimensional surface rendering (right panel). Hair follicles are demarcated by dotted lines. The closest distance between dermal ILC2 and blood vessel was measured and plotted in the right panel. **c** Image obtained from an IVM of the ear of *RS: ROSA26-LSL-tdTomato*

reporter mouse infected intradermally with $1 \times 10^3$ LmSd-GFP parasites and treated with either TSLP-neutralizing Abs or control IgG for 9 days. *RS: ROSA26-LSL-tdTomato* mice were injected with Cy5-Manocept™ to label MR^high dermal TRMs and efluor450 anti-CD31 Abs for blood vessels. Yellow-colored "R5 track" shows the paths followed by tdTomato⁺ ILC2s. Both the number and % surface of contacts per timepoint between ILC2s and dermal TRMs were analyzed and plotted in right panels. ****$P < 0.0001$ by two-sided nonparametric Mann-Whitney test (**c**). Data are representative of more than four independent mouse experiments (**a**–**c**). Source data are provided as a Source Data file.

significant portion of live cells expressed tdTomato in *Ccl24-cre: ROSA26-LSL-tdTomato* mice (Fig. 6a), and its expression was restricted to dermal TRMs without spillover to other myeloid cells (Fig. 6b). Intravital imaging of the infected animals showed strong co-localization of tdTomato expression and Cy5-Manocept™ labeling of perivascular dermal TRMs (Fig. 6c). tdTomato penetrance in dermal TRMs was more than 75% on average (Fig. 6d).

We previously demonstrated that TRMs from various tissues also produced CCL24 upon IL4/10 stimulation ex vivo[15]. To test whether the *Ccl24-cre* recombinase targets other TRMs, tdTomato expression was analyzed in multiple tissues from naïve *Ccl24-cre: ROSA26-LSL-tdTomato* mice using flow cytometry (Fig. 6e and Supplementary Fig. 6). tdTomato expression was restricted to TRMs from kidney, peritoneal cavity, adipose tissue, and liver. By contrast, tdTomato expression was not detected in lung and brain parenchymal macrophages, SiglecF⁺ alveolar macrophages or CD45.2^int microglia, while non-parenchymal or interstitial macrophages showed strong expression. tdTomato penetrance in TRMs varied from 10 to 64%, depending on the tissue. Histological analysis of these tissues corroborated the exclusive targeting of *Ccl24-cre* to various TRMs, a large fraction of which were closely associated with blood vessels and serous membranes, or to the meninges in case of the brain (Fig. 6f–j). Two groups of tdTomato⁺ TRMs co-staining with Cy5-Manocept™ were identified in the lung, one localized to the periphery along the visceral pleura (Fig. 6f; top), and the other to the interstitium surrounding the bronchus and in the perivascular space (Fig. 6f; bottom). The renal cortex was well distinguished from the medulla by Cy5-Manocept™ filtered in the collecting tubules (Fig. 6g; top). The densely packed capillaries, vasa recta in the inner medulla contained many tdTomato⁺ TRMs. In the cortex, perivascular tdTomato⁺ TRMs were associated with glomerulus and large artery (Fig. 6g; bottom). Confocal imaging also revealed tdTomato⁺

cells tightly associated with penetrating vessels in the brain parenchyma (Fig. 6h; top) as well as with CD31⁺ vessels in the pia mater of the meninges (Fig. 6h; bottom). Similar to lung, two groups of tdTomato⁺ Cy5-Manocept™⁺ TRMs in visceral adipose depot were found, one associated with mesothelial serosa, and the other with adipose capillaries (Fig. 6i). Kupffer cells within the lumen of the liver sinusoid strongly expressed tdTomato (Fig. 6j).

### TSLP from dermal TRMs mediates non-healing *L. major* infection
The *Ccl24-cre: Tslp^{f/f}* mice, along with control *Ccl24-cre: Tslp^{f/+}* and WT mice, were challenged with a high dose of LmSd. At day 12 p.i., dermal TRMs from *Ccl24-cre: Tslp^{f/f}* mice exhibited a significant loss *Tslp* expression (Fig. 7a) and these mice also had a reduction in the number of ILC2s, eosinophils, and dermal TRMs, whereas they showed increased infiltration of Ly6C^high inflammatory monocytes (Fig. 7b). IL-5 and IL-13 production from ILC2 were also decreased in *Ccl24-cre: Tslp^{f/f}* mice (Fig. 7c). Following low dose challenge, the *Ccl24-cre: Tslp^{f/f}* mice ameliorated lesion progression (Fig. 7d) and controlled parasite burdens with a 30-fold reduction compared with WT and *Ccl24-cre: Tslp^{f/+}* mice at 12 weeks post-infection (Fig. 7e).

### Human dermal macrophages are the major source of Tslp and Ccl24
The LmSd strain was originally isolated from a patient with a chronic lesion and persisting organisms[31]. Thus, it is possible that the type 2 circuitries that are regulated by dermal macrophages in murine skin to maintain their numbers and M2-like functionality might reflect a similar mechanism promoting chronic infection in humans. As transcriptional datasets from patients with cutaneous leishmaniasis are not available, we analyzed external scRNA-seq datasets from healthy and inflamed skin from adults with atopic dermatitis and psoriasis[32]. Each cluster was annotated as previously reported (Supplementary Fig. 7A,

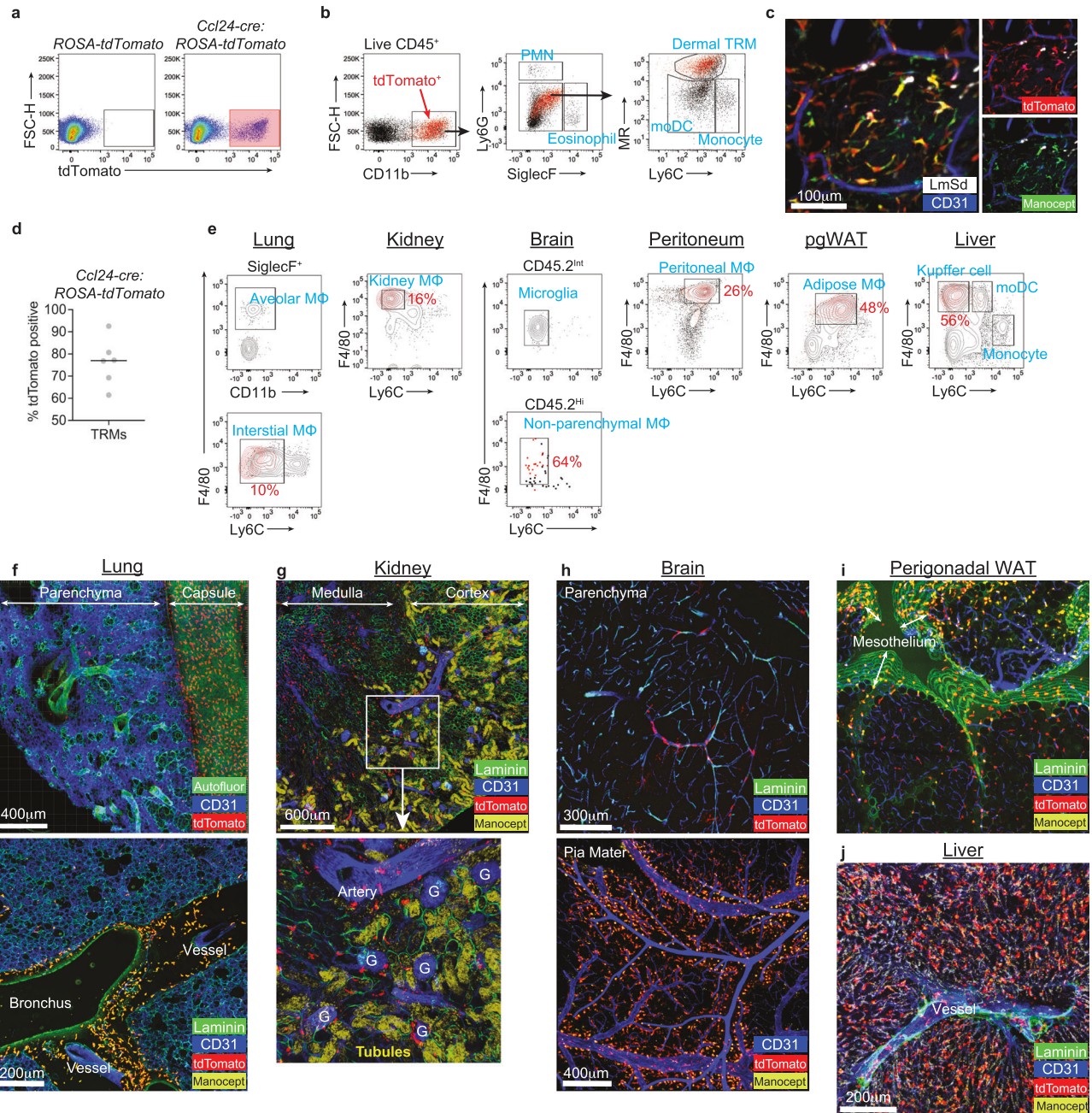

**Fig. 6 | *Ccl24-cre* transgenic mice target MHCII⁻MR^high dermal TRMs and TRMs in multiple tissues. a** Representative flow cytometric analysis of ear isolates from *ROSA26-LSL-tdTomato* mice vs. *Ccl24-cre*: *ROSA26-LSL-tdTomato* mice infected with LmSd-GFP for 8 days. **b** Live CD45.2⁺ singlets were gated into eosinophils, PMNs, inflammatory monocytes, moDCs, and dermal TRMs. The tdTomato⁺ cells in red were overlayed onto gatings. **c** Representative confocal image of ear dermis of infected *Ccl24-cre*: *ROSA26-LSL-tdTomato* mice which were intravenously injected with efluor450 anti-CD31 Abs and Cy5-Manocept™. **d** Quantification of percent tdTomato positive of dermal TRMs in (**b**) (*n* = 6). **e** Flow cytometric analysis of tdTomato⁺ cells in *Ccl24-cre*: *ROSA26-LSL-tdTomato* mice which were overlayed

onto gatings for indicated organs. pgWAT; perigonadal white adipose tissue. The complete gating strategies are shown in Supplementary Fig. 6. Percent tdTomato positive of TRMs are labeled in red. **f–j** Immunohistochemistry analysis of Alexa488-Laminin Abs stained tissue sections or whole mount from *Ccl24-cre*: *ROSA26-LSL-tdTomato* mice which were intravenously injected with efluor450 anti-CD31 Abs and Cy5-Manocept™. G glomerulus. Tubules; Renal tubules which were visualized with filtered Cy5-Manocept™. Data are representative of more than three independent mouse experiments (**a–c**). See also Supplementary Fig. 5 and 6. Source data are provided as a Source Data file.

C). As reported in allergic skin diseases[33], we detected strong *Tslp* expression in keratinocyte (Supplementary Fig. 7B). *Mrc1* was expressed most strongly by macrophage_1 and 2 and to a lesser extent by other myeloid cells, including dendritic cells, monocytes, and their derivatives (Supplementary Fig. 7C, D). Critically, we found that macrophage_1 and 2 co-expressed *Tslp* and *Ccl24* under both steady state and inflammatory conditions (Supplementary Fig. 7E).

## Discussion

This study advances our understanding of the localized immune circuitries that maintain dermal TRMs as a replicative niche for *L. major* during cutaneous infection. We have previously shown using *eoCre*: *IL4/13^f/f* mice that eosinophils provide an essential source of IL-4 to maintain both the number and M2-activation state of the dermal TRMs during *L. major* infection, and that the eosinophil chemoattractant

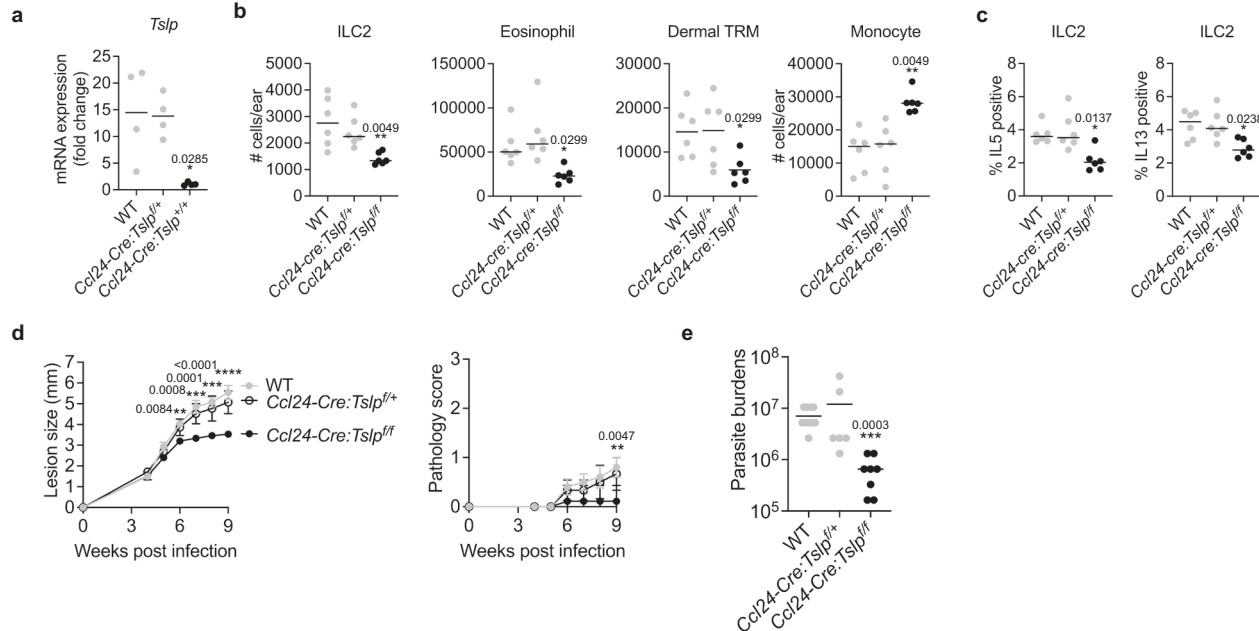

**Fig. 7 | TSLP from dermal TRMs promotes non-healing *L. major* infection.**
**a** Quantification of *Tslp* expression from the sorted MR^high dermal TRMs of WT, *Ccl24-cre: Tslp^{f/+}*, *Ccl24-cre: Tslp^{f/f}* animals at 12 days p.i. with 2 × 10^5 parasites (*n* = 4). **b** The absolute numbers of indicated cells and (**c**) % IL-5^+ or IL-13^+ ILC2s recovered from ears of WT, *Ccl24-cre: Tslp^{f/+}*, *Ccl24-cre: Tslp^{f/f}* animals were measured at 12 days p.i. with 2 × 10^5 LmSd (*n* = 6). **d** Lesion development and pathology scores over the course of infection with 10^3 LmSd metacyclic promastigotes in the ear dermis of WT (*n* = 10), *Ccl24-cre: Tslp^{f/+}* (*n* = 6), *Ccl24-cre: Tslp^{f/f}* animals (*n* = 18). **e** Parasite burdens were quantified at 9 weeks p.i. in the ear dermis of WT (*n* = 10), *Ccl24-cre: Tslp^{f/+}* (*n* = 6), *Ccl24-cre: Tslp^{f/f}* animals (*n* = 8). Values represent mean ± standard deviation. *\*P < 0.05, \*\*P < 0.01, \*\*\*P < 0.001, and \*\*\*\*P < 0.0001 by one-way ANOVA with Dunn's post-test compared to *Ccl24-cre: Tslp^{f/f}* animals (**a**–**e**). Data are representative of two independent experiments (**a**–**e**). Source data are provided as a Source Data file.

CCL24 was required for the tissue accumulation of eosinophils and their co-localization with dermal TRMs in the lesion[7,15]. As CCL24 was produced by the dermal TRMs themselves, the current studies unveil an additional layer of TRM self-maintenance involving their production of TSLP, which was required to maintain the number of IL-5^+ ILC2s in the site of infection. Thus, the selective deletion of IL-5^+ ILC2s in the *R5: Gata3^{f/f}* mice, or the conditional ablation of *Tslp* from dermal TRMs in the *Ccl24-cre: Tslp^{f/f}* mice, in each case resulted in decreased numbers of ILC2s, eosinophils and dermal TRMs in the lesions, and ameliorated the infection outcome.

TSLP has been previously identified as one of three principal alarmins produced by epithelial cells and stromal cells in response to insults at barrier surfaces[33]. In the skin, TSLP production by epidermal keratinocytes can be induced by mechanical injury, environmental allergens, and cutaneous pathogens such as *Staphylococcus aureus*[23]. Using intradermal inoculation by needle to mimic the site of *Leishmania* delivery by sand flies, we show that dermal TRMs produce TSLP, which is critical to mediate non-healing cutaneous infection. Although keratinocytes are a well-known source of all three alarmins[22], we failed to detect IL-33, IL-25, or TSLP expression in the scRNA-seq epithelial cluster at 12 days post-infection, suggesting that there was minimal disturbance of the epithelial barrier in our murine infection model. Previously, TRMs as a source of infection driven alarmins has only been described in the context IL-33 production by alveolar macrophages in response to respiratory viruses[34,35]. The release and function of TSLP by dermal TRMs during *L. major* infection appears to be non-redundant, as neither IL-33 nor IL-25 transcripts were detected in the dermal TRM cluster, and mice deficient in these cytokines did not alter their infection outcome. Interestingly, the ability of TSLP to promote inflammation in a murine model of atopic dermatitis was also found to be independent of either IL-33 or IL-25, although the source of TSLP was not addressed[20]. Nonetheless, the 10-fold reduction in parasite burdens observed in the *Tslpr^{−/−}* and *Ccl24-cre: Tslp^{f/f}* mice in comparison to the WT controls, did not approach the nearly 100-fold

reduction observed in IL-5^+ ILC2 deficient *R5: Gata3^{f/f}* mice. This suggests a role for additional alarmin(s), possibly IL-18, that may activate a major subset of skin ILC2s, known as IL18R1-expressing ILC2s[36].

The function of TSLP from dermal TRMs in the setting of the *L. major* infected dermis appears to be related to its ability to locally activate ILC2s, as both the absolute number and the frequency of ILC2s producing IL-5/IL-13 in the lesion were significantly reduced in the infected *Ccl24-cre: Tslp^{f/f}* mice. Using the *R5: Gata^{f/f}* mice to directly target IL5^+ ILC2 for deletion, we could show that these cells were required for the evolution of the non-healing response. So far as we are aware, this is the first description of ILC2s impacting the infection outcome in leishmaniasis. An association of ILC2s with clinical disease has been reported in patients with diffuse cutaneous leishmaniasis (DCL), who had increased frequencies of circulating ILC2s as compared to patients with localized lesions or mucosal disease, although the number of patients in each group was small[37].

ILC2s in peripheral tissue are important regulators of eosinophil survival and recruitment by virtue of their constitutive expression of IL-5 and its upregulation during type 2 inflammation[24]. ILC2-derived IL-5 is implicated in the homeostatic homing of eosinophils into the small intestine and visceral adipose tissue, and in the accumulation of eosinophils during allergic inflammation in the lungs[17,38]. In the skin, ILC2-derived IL-5 is implicated in the strong eosinophil infiltrate observed in an IL-33 transgene-driven model of atopic dermatitis in mice[39]. In the current studies, targeting IL-5^+ ILC2s for deletion in the *R5: ROSA26iDTR* or *R5: Gata^{f/f}* mice led to the reduced number of eosinophils, with downstream effects on the number of dermal TRMs due to the loss of a critical source of IL-4 required for their maintenance. Given the reported interplay between ILC2s and alternatively activated macrophages in various contexts such as parasite infections, metabolic regulation, and tissue repair[40], IL-5^+ ILC2s may exert a direct influence on the number and M2-like phenotype of dermal TRMs by secreting type 2 cytokines. Our observations also indicate that antibody-mediated neutralization of TSLP leads to a reduction in the

number of ILC2s and their interactions with dermal TRMs in infected skin. Since both cell types are localized within the perivascular niche in the dermis, it is possible that their close proximity and cross-regulation is facilitated by the direct release of TSLP by dermal TRMs. Interestingly, the co-localization of ILC2s with adventitial stromal cells (ASCs) around lung bronchi and larger vessels has been previously attributed to the production of IL-33 and TSLP by ASCs, which supports the accumulation and function of ILC2s in those settings[41].

A recent report confirmed eosinophils as the main intra-lesional source of IL-4 in *L. major* infected, C57BL/6 mice[42]. Evidence that IL-4/13 from eosinophils contributed to the upregulation of *Tslp* expression in dermal TRMs was provided by the scRNA-seq comparisons of relative *Tslp* expression within the dermal TRM cluster from infected, wild type and *eoCre: IL4/13^(f/f)* mice. A prior in vitro study showed that exogenous IL-4 could enhance the ability of LPS stimulated primary human lung macrophages to release TSLP[43]. The reduced frequency of IL-5$^+$/IL-13$^+$ ILC2s in the *eoCre: IL4/13^(f/f)* mice is likely explained by the reduced level of TSLP provided by the dermal TRMs, although we cannot rule out a direct, bi-directional interaction between the eosinophils and ILC2s. Indeed, ILC2s express IL-4Rα, and IL-4 from eosinophils or basophils has been shown to promote ILC2 activation in inflamed lung or skin[44,45]. As TSLP has been shown to act on pulmonary macrophages directly to amplify their alternative activation state[43,46], and as the TRM cluster expresses TSLP receptor (*Crlf2*), we also cannot rule out the possibility that TSLP functions in an autocrine manner and cooperates with IL-4 to maintain the M2-like phenotype of the dermal TRMs during infection. Evidence that *Tslp* and *Ccl24* expressions are integral components of the M2-like activation program of dermal TRMs in the mouse was extended to a human scRNA-seq dataset in which the cells in the macrophage cluster marked by expression of *Mrc1* co-expressed both *Tslp* and *Ccl24* in healthy and allergic skin biopsies (33). High expression of *Tslp* in the skin of patients with diffuse cutaneous systemic sclerosis was also detected in perivascular CD163$^+$ macrophages[47]. We also observed an increase in the number of inflammatory monocytes/moDCs in *R5: ROSA26iDTR* or *R5: Gata^(f/f)* mice, as well as in *Ccl24-cre: Tslp^(f/f)* mice, which aligns with our previous findings in *eoCre: IL4/13^(f/f)* mice[15]. This suggests that TSLP/CCL24 originating from dermal TRMs has direct or indirect effects on monocyte recruitment, perhaps by creating an open resident macrophage niche. We have previously reported elevated levels of CCL3/4 (macrophage inflammatory protein 1α, MIP1α, and MIP1β) in dermal TRMs from *eoCre: IL4/13^(f/f)* mice, which are known chemoattractants for monocytes[15]. Recruited monocytes and monocyte-derived cells likely contribute to the control of *L. major* infection, as in contrast to the dermal TRMs, these cells show up-regulated expression of inducible nitric oxide synthase[7,48,49].

The subpopulations of MHCII$^-$ and MHCII$^+$ dermal TRMs in the mouse have been distinguished previously with respect to their DEGs in normal skin and the extent and rate of their replacement by monocyte-derived cells, with the MHCII$^-$ TRMs replaced only partially[5,6]. Here, we further define the distinct transcriptional programs of these two subpopulations of dermal TRMs in inflamed skin. The MHCII$^-$ TRMs showed higher expression of genes involved in alternative activation, fatty acid oxidation, and wound healing, while MHCII$^+$ dermal TRMs showed higher expression of genes associated with monocytic markers, pro-inflammatory response, and antigen presentation. The expression of *Mrc1*, *Ccl24*, and *Tslp* were all strongly associated at the single cell level, particularly within the MHCII$^-$ population. Interestingly, the hyaluronan receptor 1 (*Lyve1*), which was also highly expressed in MHCII$^-$ dermal TRMs, has been used to identify two subsets of interstitial macrophages (Lyve1$^{low}$MHCII$^{high}$ vs Lyve1$^{high}$MHCII$^{low}$) that exist across tissues and exhibit different functions and sub-organ localization (nerve- vs vessel-associated)[30]. We and others have described the perivascular localization of MHCII$^-$MR$^{high}$ dermal TRMs[7,50]. Nerve-associated CX$_3$CR1$^{high}$ skin macrophages were

shown to be involved in sensory neuron regeneration after injury[51]. The role of Lyve1$^{low}$MHCII$^+$MR$^{low}$ TRMs during cutaneous infection remains to be addressed, particularly with regard to nerve-macrophage interactions.

Although cre transgenic animals that permit selective visualization or depletion of TRMs in the liver or brain have been recently developed[52,53], a cre transgenic system that targets TRMs across multiple tissues has not been described, so far as we are aware[54]. Sequential double promoter systems, such as MM$^{DTR}$ (*Lysm^(Cre)* x *Csf1r^(LSL-DTR)*) and *Cx3cr1^(CreER/+)* x *Csf1r^(Flox/Flox)* mice, delineated resident macrophages in multiple tissues but also targeted monocytes and other myeloid cells[55,56]. A binary cre transgenic system with expression driven by two distinct promoters, *Lyve1^(ncre)* with *Cx_3cr1^(ccre)*, and crossed with mice harboring a *R26/CAG-tdTomato* reporter allele, targeted perivascular TRMs in the brain and other tissues[53]. The main drawback, however, was the low efficiency of reconstitution of the functional enzyme complex, resulting in only 5% penetrance of tdTomato expression in the perivascular TRMs. Here, we established a single *Ccl24* promoter-driven Cre approach to visualize dermal TRMs in both steady and infectious states. Flow cytometric analysis of *Ccl24-cre: ROSA26-LSL-tdTomato* mice revealed around 80% of MHCII$^-$MR$^{high}$ dermal TRMs during *L. major* infection were tdTomato$^+$ while all other skin myeloid populations remained negative. Strikingly, the naïve *Ccl24-cre: ROSA26-LSL-tdTomato* mice permitted visualization of TRMs from most tissues in steady state condition. The labeled TRMs largely occupied two distinct sub-tissular niches. One population was perivascular, including lung interstitial TRMs, kidney TRMs associated with vasa recta and glomerulus, brain TRMs, adipose TRMs, and Kupffer cells in liver sinusoids. Of note, the labeled TRMs in the brain appeared identical to those labeled in the *Lyve1^(ncre): Cx_3cr1^(ccre):R26-tdTomato* mice previously reported[53], associated with both highly vascularized pia mater of the meninges and penetrating vessels in the parenchyma. This result agrees with our finding that perivascular MHCII$^-$MR$^{high}$ TRMs highly express both *Ccl24* and *Lyve1* in the dermis. The *Ccl24* expression by these TRMs in physiological homeostasis across different tissues likely reflects a shared component of their alternative activation states. The other labeled TRMs in the naïve, *Ccl24-cre: ROSA26-LSL-tdTomato* mice were sheltered in tissue-linings, such as serosal membranes of lung and visceral adipose tissue. These TRMs might play a tissue-protective role by sequestering inflammatory responses, as was described for the barrier forming layer of synovial macrophages which restrains inflammation around joints, or the "cloaking" function of interstitial macrophages which limits the tissue damage around microlesions[57,58].

In summary, we demonstrate that dermal TRMs orchestrate localized type 2 circuitries with ILC2s and eosinophils, mediated by TSLP and CCL24 respectively, to promote non-healing cutaneous leishmaniasis. We identified the dermal TRMs themselves as an essential source of TSLP to self-maintain both their number and M2-like activation program within the strong T$_H$1 immune environment of the *L. major* infected dermis.

## Methods
### Ethical statement
Our research complies with all relevant ethical regulations. All the mice used in these studies were used under a study protocol approved by the NIAID Animal Care and Use Committee (protocol number LPD 68E). All aspects of the use of animals in this research were monitored for compliance with The Animal Welfare Act, the PHS Policy, the U.S. Government Principles for the Utilization and Care of Vertebrate Animals Used in Testing, Research, and Training, and the NIH Guide for the Care and Use of Laboratory Animals.

### Mice
C57BL/6NTAC mice and *Il25^(-/-)* mice were obtained through a supply contract between the National Institute of Allergy and Infectious

Diseases (NIAID) and Taconic Farms. *Il33*$^{-/-}$ animals were generated by cross-breeding B6(129S4)-Il33tm1.1Bryc/J (*Il33*$^{f/f}$) mice and B6.C-Tg(CMV-cre)1Cgn/J (*CMV-Cre*) mice, both of which were purchased from The Jackson Laboratory. B6(C)-*Il5*$^{tm1.1(icre)Lky}$/J (also known as *R5*) mice, *ROSA26-LSL-tdTomato* (also known as *Ai14*) mice, C57BL/6-*Gt(ROSA)26Sort*$^{tm1(HBEGF)}$ $^{Awai}$/J mice (*ROSA26iDTR*) mice, and *Gata3*tm1.1Mbu/J (*Gata3*$^{f/f}$) mice were purchased from The Jackson Laboratory. Dr. Warren J Leonard (NHLBI) kindly provided C57BL/6 *Tslpr*$^{-/-}$ mice. *Il4*$^{-/-}$ and *eoCre: Il4/13*$^{f/f}$ mice on a C57BL/6 background have been described previously[15]. All the mice used in these studies were female, 6–8 weeks old, and were bred and maintained in the NIAID animal care facility under specific pathogen-free condition and at a constant cycle of 14 h in the light (<300 lux) and 10 h in the dark. Colonies were maintained at 20–24 °C and 40–60% humidity, with free access to food and water.

## Generation of Ccl24-Cre and Tslp^f/f mice

*Ccl24-Cre and Tslp*$^{f/f}$ mice were generated *via* CRISPR/Cas9-mediated insertion in collaboration with Mouse Genetics and Gene Modification Section, Comparative Medicine Branch, NIH. Guide sites were selected using CRISPOR design tools (http://crispor.tefor.net/), and CRISPR/Cas9 guide RNA (synthetic gRNA) was manufactured by Synthego (Redwood City, CA). Cas9 nuclease was purchased from IDTdna (Coralville, IA). Targeting strategies and gRNA sequences are displayed in Supplementary Fig. 5. To generate the plasmids for Homology Directed Repair (HDR) templates, IRES-Cre sequence was cloned from Plasmid B108-Otx2-IRES-Cre (#73993, Addgene) and inserted into pUC19 between 0.8 kb homology arms to generate *Ccl24-IRES-Cre* donor construct (pUC19-ccl24-IRES-Cre) using Gibson assembly method (NEBuilder Hifi DNA Assembly Cloning Kit, New England Biolabs). One PAM-blocking silent mutation, ccg (Proline) to cgg (Arginine) was introduced to prevent re-editing by Cas9 after homology directed repair. For *Tslp-loxp* donor construct, both-ends 1 kb homology arms and floxed region containing exon 1 and 2 were assembled into pUC19 backbone. Overlapping forward primers to generate both floxed and 3′-homology arm DNA fragments for Gibson assembly were designed to include 34bp-long LoxP sequences. For microinjection, the C57BL/6Tac females were injected with 7.5IU of pregnant mare serum gonadotropin (Prospec, Israel) followed by 7.5IU of human chorionic gonadotropin (Sigma, St. Louis, MO) after 47 h, then mated with C57BL/6Tac males. The following morning, fertilized one-cell-stage embryos were collected, washed 6 times with EmbryoMax M2 media (Millipore Sigma, Burlington, MA), and then cultured in EmbryoMax KSOM media (Millipore Sigma, Burlington, MA) at 6% CO$_2$, 37 °C for 3 h or until the microinjection. The embryos were transferred to 50 mm glass bottom dish (Mattek, Ashland, MA) in EmbryoMax M2 drop, overlaid with embryo-tested mineral oil (Sigma, St Louis, MO), and microinjected using an DMi8 inverted microscope (Leica, Wetzlar, Germany) equipped with a set of TrasferMan4 micromanipulator and FemtoJet4i (Eppendorf, Hamburg, Germany). Microinjection mixture was prepared fresh as following: 10 ng/µl of HiFi Cas9 protein, 5 ng/µl of sgRNA, 5 ng/µl of plasmid in microinjection buffer (10 mM Tris-HCL pH7.5 with 0.25 mM EDTA), backfilled to microinjection needles for pronuclear injection. Injected embryos were implanted into the oviducts of pseudo-pregnant surrogate CD-1 females (Charles River Labs, Wilmington, MA) after 1 h culture in EmbryoMax KSOM at 6% CO$_2$, 37 °C. Offspring born to the foster mothers were genotyped by PCR. The primer sequences for genotyping were as follows: ccl24 for, TGTGGAGACCAGAGGCTAAT; IRES rev, GCATTCCTTTGGCGAGAG; tslp for, TGACTCCAGTCTGTGCTTTC; tslp 5flox rev, AGCATACATTATACGAAGTTATCATCA; tslp 3flox for, AATGTATGCTATACGAAGTTATTGCT; tslp rev, TTGAGGGCTTCTCTTGTTCTC.

## In vivo injections

Cy5-Manocept™ (Navidea Biopharmaceuticals) is a fluorescently labeled derivative of FDA-approved 99MTC-Tilmanocept™ targeting MR. Naïve mice were injected with 25 µg Manocept™ intravenously with 20 µg anti-CD31 Abs (Clone#390, Invitrogen) in total volume not to exceed 100uL per mouse during IVM. For TSLP neutralization in vivo, animals were intraperitoneally injected with 300 µg of TSLP Abs (Clone# MAB555, R&D systems) at day 0, 2, and 4 post-infection. To deplete ILC2s in *R5: ROSA26iDTR* mice, 20 ng DT was administered intradermally at day 0 followed by 200 ng DT intraperitoneal injection at day 2 and 4 post-infection.

## Adoptive transfer of ILC2

To obtain a sufficient number of ILC2s for adoptive transfer, we employed a previously described method[59]. In brief, WT animals on a C57BL/6 background were administered a daily injection of 1 µg of recombinant mouse interleukin-25 (rmIL-25, R&D Systems) for 1 week. After completing the treatment regimen, we harvested spleens and performed immunomagnetic enrichment of ILC2s using the Easy-SepTM Mouse ILC2 Enrichment Kit (STEMCELLTM Technologies). These enriched ILC2s were subsequently FACS-sorted (as Lineage⁻CD45⁺IL17RB⁺ICOS+IL7ra$^{int}$) and expanded in vitro in the presence of IL-7 and IL-33 (10 ng/ml each). On day 0, we intravenously injected $0.5 \times 10^6$ in vitro expanded ILC2s.

## *Leishmania major* infection

The *L. major* Seidman strain (MHOM/SN/74/SD) (LmSd) was maintained as follows: promastigotes were grown at 26 °C in medium 199 supplemented with 20% heat-inactivated FCS (Gemini Bio-Products), 100 U/mL penicillin, 100 µg/mL streptomycin, 2 mM L-glutamine, 40 mM Hepes, 0.1 mM adenine (in 50 mM Hepes), 5 mg/mL hemin (in 50% triethanolamine), and 1 mg/mL 6-biotin (M199/S). Parasites expressing a green fluorescent protein (LmSd-GFP) were grown using the identical culture medium. Infective-stage, metacyclic promastigotes were isolated from stationary cultures (5–6 days) by density gradient centrifugation, as described previously[60]. Mice were then inoculated with metacyclic promastigotes in the ear dermis by intradermal injection in a volume of 10 µL. Lesion development was monitored weekly by measuring the diameter of the ear nodule with a direct-reading Vernier caliper (Thomas Scientific). Lesion pathology was also evaluated and scored as follows: 0 = no ulceration, 1 = ulcer, 2 = half ear eroded, 3 = ear completely eroded.

## Processing of ear tissues and evaluation of *L. major* parasite burden

Ear tissues were prepared as previously described[61]. Briefly, the two sheets of infected ear dermis were separated, deposited in DMEM containing 0.2 mg/mL Liberase TL purified enzyme blend (Roche Diagnostics Corp.), and incubated for 1.5 h at 37 °C. Digested tissues were processed in a tissue homogenizer for 3.5 min (Medimachine; Becton Dickinson) and filtered through a 70 µm cell strainer (Falcon Products) to obtain single cell suspension. Parasite titrations were performed as previously described[62]. Briefly, tissue homogenates were serially diluted in 96-well flat-bottom microtiter plates containing 100 µL M199/S. The number of viable parasites in each ear was determined from the highest dilution at which promastigotes could be grown out after 7–10 days of incubation at 26 °C.

## Immunolabeling and flow cytometry analysis

Single-cell suspensions were stained with LIVE/DEAD Fixable Aqua Dead Cell Stain Kit (Thermofisher) and incubated with an anti-Fcγ II/III (CD16/32) receptor Ab (2.4G2, BD Biosciences) in PBS containing 1% FCS followed by fluorochrome-conjugated antibodies for 1 h on ice. The following antibodies (1:200 dilution) were used for surface staining: PE anti-mouse CD90.2 (Thy-1.2, Cat#140307, Biolegend); PE/Cy7 anti-mouse CD2 (RM2-5, Cat#100113, Biolegend); APC anti-mouse CD3 (17A2, Cat#100235, Biolegend); Alexa488 anti-mouse TCRβ (H57-597, Cat#109216, Biolegend); PerCP/Cy5.5 anti-mouse TCRγ/δ (GL3,

Cat#118117, Biolegend); FITC, APC/Cy7, and Brilliant Violet™ 421 anti-mouse Ly6G (1A8, Cat#127605/127623/127627, Biolegend); APC/Cy7 anti-mouse Ly6C (HK1.4, Cat#128025, Biolegend); FITC and PE/Cy7 anti-mouse CD11b (M1/70, Cat#101205/101215, Biolegend); PerCP/Cy5.5, Brilliant Violet™ 421, and APC/Cy7 anti-mouse NK1.1 (PK136, Cat#108727/108731/108723, Biolegend); Alexa647 and APC anti-mouse CD206 (C068C2, Cat#141711/141707, Biolegend); PE and Brilliant Violet™ 421 anti-mouse Siglec-F (s17007L, Cat#155505/155509, Biolegend); PerCP/Cy5.5 anti-mouse CD45.2 (104, Cat#109827, Biolegend); APC and APC/Cy7 anti-mouse F4/80 (BM8, Cat#123115/123117, Biolegend). For intracellular detection of cytokines, single cell suspension harvested from tissues were cultured in the presence of PMA/Ionomycin and Brefeldin A (Biolegend) for 4 h at 37 °C. The staining of surface and cytoplasmic cytokines/chemokine was performed sequentially. Cells were first incubated with LIVE/DEAD Fixable Aqua Dead Cell Stain Kit (Thermofisher). They were stained for their surface markers, then fixed and permeabilized using BD Cytofix/Cytoperm (BD Biosciences), and finally stained for detection of cytokines/chemokine for 30 min on ice. The following antibodies (1:200 dilution) were used for intracellular staining: Brilliant Violet™ 421 anti-mouse IL-5 (TRFK5, Cat#504311, Biolegend); Alexa488 anti-mouse IL-13 (eBio13A, Cat#12-7133-41, Thermo-Fisher Scientific). The data were collected using FacsDIVA software and a FacsCANTO II flow cytometer (BD Biosciences) and analyzed with FlowJo software (Tree Star).

## Intravital microscopy

*R5: ROSA26-LSL-tdTomato* and *Ccl24-cre: ROSA26-LSL-tdTomato* mice were intravenously injected with 20 μg of eFluor450 anti-mouse CD31 (390, Cat#48-0311-82, Invitrogen) to outline blood vessels, and with 25 μg Cy5-Manocept™ to visualize MR⁺ TRMs, immediately prior to imaging. Non-invasive intravital imaging of mouse ear was performed using Leica Deep In Vivo Explorer (DIVE) inverted confocal microscope (Leica Microsystems) with dual multiphoton lasers Mai Tai Deep See and InSight Deep See (Spectra Physics), and full range of visible lasers. Additionally, the microscope was equipped with external and internal hybrid detectors; L x 25.0 water-immersion objective with 0.95 NA (Leica Microsystems) and 2 mm working distance; a motorized stage; and Environmental Chamber (NIH Division of Scientific Equipment and Instrumentation Services) to maintain 37 °C. Anesthesia was induced with 2% Isoflurane (Baxter) and maintained at 1.5% during imaging. A temperature sensor was positioned on the stage near the animal. Mai Tai was tuned to 880 nm excitation; InSight was tuned to 1150 nm. Diode laser was used for 405 nm excitation; Argon laser for 488 nm excitation; DPSS laser for 561 nm excitation; and HeNe laser for 633 nm excitation wavelengths. All lasers were tuned to minimal power (between 1 and 5%). For time-lapse imaging, small tiled images of 2 × 2 fields were recorded over time, and for static images 5 × 5 fields were acquired using Tilescan application of LAS X. Z stacks consisting of 6−8 single planes (3−5 μm each over a total tissue depth of 50−70 μm) were acquired every 45 s for a total observation time between 1 to 6 h for 4D reconstruction, surface modeling and tracking with the Imaris software (Imaris version 9.9.1, Bitplane AG, Zurich, Switzerland). Cells were segmented as 3D surface model and tracked using modified Imaris autoregressive tracking algorithm. Cell tracks are shown as cylinders. Contacts between cells were calculated using "Kiss and Run Analysis" XTension for Imaris. Distance Transformation XTension of the Imaris software outside of the target surface object was used to determine closest surface-to-surface distance of the tracked objects.

## Tissue sections

Confocal microscopy of live tissue sections was performed for analysis of tissue architecture, cell segregation, and cell numbers in mouse lung, kidney, brain, fat, and liver ex vivo, at their physiological conditions. After euthanasia, mouse lungs were inflated with 1.5% of low-melt agarose in RPMI at 37 °C. Inflated tissues were kept on ice, in 1% FBS in PBS, and sliced into 300−350 μm sections using Leica VT1000 S Vibrating Blade Microtome (Leica Microsystems). Mouse kidney, brain, and liver were embedded in 1.5% agarose in RPMI at 37 °C prior to sectioning. Fat tissue was imaged directly. Tissue sections were stained with AF488 anti-mouse Laminin (1:200 dilution, NB300-144AF488, Novus Biologicals) for 2 h on ice. After staining, sections were washed three times and cultured in complete lymphocyte medium (Phenol Red-free RPMI supplemented with 10% FBS, 25 mM HEPES, 50 μM β-ME, 1% Pen/Strep/L-Glu and 1% Sodium Pyruvate) in humidified incubator at 37 °C. Sections were held down with tissue anchors (Warner Instruments) in 14 mm microwell dishes (MatTek), and imaged using DIVE microscope.

## Single-cell RNA sequencing (scRNA-seq) and CITE-seq sequencing

Single cell suspension and antibody staining were performed as described above except for addition of 100 unit/ml SUPERase•InTM RNase Inhibitor (ThermoFisher) to all of the tissue-digestion/staining buffers to inhibit the strong ribonuclease activity in murine skin. The following antibodies were used for surface staining (1:200 dilution) and for Chromium Single Cell 3′ Feature Barcode library (1:40 dilution): PE anti-mouse CD45.2 (104, Cat#109808, Biolegend); TotalSeq™-B0182 anti-mouse CD3 (17A2, Cat#100257, Biolegend); TotalSeq™-B0001 anti-mouse CD4 (RM4-5, Cat#100573, Biolegend); TotalSeq™-B0014 anti-mouse CD11b (M1/70, Cat#101273, Biolegend); TotalSeq™-B0106 anti-mouse CD11c (N418, Cat#117359, Biolegend); TotalSeq™-B0093 anti-mouse CD19 (6D5, Cat#115563, Biolegend); TotalSeq™-B0118 anti-mouse NK1.1 (PK136, Cat#108763, Biolegend); TotalSeq™-B0015 anti-mouse Ly6G (1A8, Cat#127659, Biolegend); TotalSeq™-B0012 anti-mouse c-Kit (2B8, Cat#105849, Biolegend); TotalSeq™-B0847 anti-mouse ICOS (7E.17G9, Cat#117425, Biolegend); TotalSeq™-B0114 anti-mouse F4/80 (BM8, Cat#123155, Biolegend); TotalSeq™-B0157 anti-mouse CD45.2 (104, Cat#109859, Biolegend); TotalSeq™-B0115 anti-mouse FcεR1α (MAR-1, Cat#134341, Biolegend); TotalSeq™-B0002 anti-mouse CD8α (53-6.7, Cat#100783, Biolegend); TotalSeq™-B0431 anti-mouse SiglecF (S17007L, Cat#155517, Biolegend); TotalSeq™-B0810 anti-mouse CD138 (281-2, Cat#142536, Biolegend); TotalSeq™-B0120 anti-mouse TCRβ (H57-597, Cat#109261, Biolegend); TotalSeq™-B0211 anti-mouse TCRγδ2 (UC3-10A6, Cat#137715, Biolegend); TotalSeq™-B0301 to 0304 anti-mouse Hashtag 1 to 4 (M1/42 and 30-F11, Cat#155831/155833/155835/155837, Biolegend). CD45⁺ and CD45⁻ cells were sorted and mixed in 9:1 ratio and processed through the Chromium Single cell 3′ v3.1 Library kit (10x Genomics) per the manufacturer's protocol. Chromium Next GEM (Gel Beads-in-emulsion) chips were loaded with 8300 sorted cells, targeting 5000 single cells for analysis from each ear. The RNAs from lysed single cells in gel beads were reversed transcribed into cDNA with Barcodes. Library fragment-size distribution was assessed using the Bioanalyzer 2100 and the DNA high-sensitivity chip (Agilent Technologies), followed by amplification, fragmentation, adapter ligation, and size selection to generate both 3′ gene expression libraries and cell surface protein libraries. Libraries were sequenced on an Illumina NovaSeq 6000 by Psomagen, Inc.

## scRNA-seq data analysis

Read alignment to the mouse genome (mm10) and generation of gene count matrices from raw sequence fasq files were performed using the CellRanger pipeline, version 5.0.1 (10x Genomics) in computational resources of the NIH High Performing Computation Biowulf cluster (http://hpc.nih.gov). The acquired gene count matrices were further processed using Seurat package, version 4.3.0[63] in R, version 4.2.1. For quality control, (i) genes expressed by less than three cells were filtered out, and (ii) the cells with lower than 200 features detected were also excluded from the matrices. Filtering based on the proportion of mitochondrial RNA to filter out apoptotic cells was intentionally

excluded to retain CD45⁻ non-hematopoietic cells, which are more prone to stress during cell isolation, in the dataset. To align shared cell populations across datasets, multiple experimental single-cell datasets were integrated using the strategy to anchor diverse datasets. First, the data were combined into a single integrated object and split by sample, then normalized by the *NormalizeData* function individually. Highly variable features (2000 features) were identified independently. Anchors, features commonly shared across datasets, were found by *FindIntegrationAnchors* function in the Seurat. Using the identified anchorage features, multiple datasets were integrated followed by scaling, clustering (resolution = 0.3) based on the defined principal components (1:30), and dimensionality reduction by UMAP for visualization. For CITE-seq (ADT, Antibody-Derived Tags, hereafter), ADT count metrices were incorporated into the established Seurat object, normalized according to the centered-log-ratio method and scaled. Cluster identities were first roughly determined by SingleR, version 1.0.5[64] with transcriptomic mouse datasets from Immgen database[27] as reference and then, by examining each cluster markers, calculated by *FindAllMarkers* function by Wilcoxon rank sum test (default). Differentially expressed genes (DEGs) of between clusters were calculated using *FindMarker* function. Gene clustering for heatmap was performed by hierarchical clustering (hclust function with "complete" or "ward.D2"). Connectivity map (cMAP) analysis[28,65] was performed using DEGs between MHCII⁺ and MHCII⁻ dermal macrophages or between dermal monocytes and MHCII⁺/⁻ dermal macrophages derived from the Gene Expression Omnibus data series GSE49358[5]. For individual cells, cMAP generated enrichment scores that quantified the degree of enrichment (or "closeness") to the given gene signatures. The enrichment scores were scaled and assigned positive or negative values to indicate enrichment for signature genes of comparing cells. A permutation test ($n$ = 1000) between gene signatures was performed on each enrichment score to determine statistical significance. To overcome the dropout effect in single cell data, we used MAGIC package, version 2.0.3[66] with the default setting (knn = 5, decay = 1). The magic transformed gene expression values were used for visualization of featureplots (Fig. 3b, c), violin plots (Fig. 3f, g) and 3D plot (Fig. 4d). The functional enrichment analyses were generated through the use of QIAGEN IPA (https://digitalinsights.qiagen.com/IPA).

Single cell RNA-seq AnnData file (submission_210120.h5ad) of human skin rashes[32] was downloaded from Zenodo (https://zenodo.org/record/4569496) and converted into a Seurat object using Scanpy Python module, version 1.9.1[67] and Seurat R package, version 4.3.0[63] in Rstudio, Version 2022.12.0. Cells were randomly down sampled to a maximum of 10,000 cells per condition (total of 59,288 cells after QC). The Seurat object was split into a list according to each condition (healthy, atopic dermatitis, or psoriasis) and processed using Seurat R package, version 4.3.0. UMI counts were normalized and scaled using *NormalizeData* and *ScaleData* functions in Seurat, and highly variable genes were identified using *FindVariableFeatures* (2000, "vst"method). Data integration was performed by selecting genes that are variable in most samples with *SelectIntegrationFeatures* and identifying cells that can be used as "anchors" with the *FindIntegrationAnchors*, to integrate the different samples using the *IntegrateData* function. After scaling the integrated data with function *ScaleData*, dimensionality reduction was performed using *RunPCA* and *RunUMAP* and clustering with *FindNeighbors* and *FindClusters* with 14 principal components and a resolution of 0.6. For the analysis of immune cells in the same dataset, the rds file (Three_together.rds)[68] was downloaded containing the Seurat R object with cell identities available on Zenodo (https://zenodo.org/record/6470377#.Yl_f2oVBxPa). *DietSeurat* function in Seurat R package, version 4.3.0 was used to extract UMI counts into a new Seurat object. Only cells derived from atopic dermatitis, psoriasis, and healthy adult donors were kept and randomly down sampled to a maximum of 10,000 cells per donor (total of 98,133 cells). The Seurat object was split into a list according to each donor (4 atopic dermatitis,

3 psoriasis, and 5 healthy skin) and processed using Seurat as described above for the full set of clusters, using 50 principal components and a resolution of 0.6. All clusters were annotated according to cell type identities as assigned in Reynolds et al. [32].

## Statistics and reproducibility
The differences in values obtained for two different groups were determined using non-parametric Mann-Whitney test. For comparisons of multiple groups, one-way analysis of variance (ANOVA) followed by Dunn's post-test was used. Analyses were performed using Prism 8.0 software (GraphPad). A minimum of two independent experiments were conducted for all the studies, yielding consistent results. No statistical method was used to predetermine the sample size. No data was excluded. The experiments were not randomized. The Investigators were not blinded to allocation during experiments and outcome assessment.

## Reporting summary
Further information on research design is available in the Nature Portfolio Reporting Summary linked to this article.

## Data availability
The source data for each graph are provided as a Source data file. Single cell RNA sequencing data that support the findings of this study have been deposited in Gene Expression Omnibus (GEO) with the GSE243853 accession number. Source data are provided with this paper.

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

## Acknowledgements

We thank Dr. Warren J Leonard (NHLBI) for provision of the *Tslpr*[−/−] mice. We thank Dr. Calvin Eigsti (NIAID) for help with the cell sorting. R code for cmap was kindly provided by Dr. Jinmiao Chen from Singapore Immunology Network (SIgN), A*STAR. This work was supported in part by the Intramural Research Program of the National Institute of Allergy and Infectious Diseases, National Institutes of Health.

## Author contributions

S.H.L. and D.L.S. designed the study; S.H.L., B.K., O.K., T.R.F., K.C. and J.K. performed and/or analyzed experiments; S.H.L., J.S.K., B.L.K. and D.L.S. reviewed the data; S.H.L. and D.L.S. wrote the manuscript.

## Competing interests

The authors declare no competing interests.
