## [Peer Review File · Nature Communications]

Dermis resident macrophages orchestrate localized ILC2 eosinophil circuitries to promote non healing cutaneous leishmaniasisREVIEWER COMMENTS

Reviewer #1 (Remarks to the Author):

In this manuscript, Lee et al report the role of TSLP and CCL24 producing, M2-like dermal tissue resident macrophages (TRM) in non-healing cutaneous lesions during *Leishmania* major infection in mice. This manuscript is very thorough and provides nice insights into macrophage heterogeneity and its importance in infectious immunity and inflammation. Of particular interest to the scientific community, the team generated *Ccl24-Cre* and *Tslp-flox* mice using CRISPR/Cas9 to specifically target this population, and showed that targeted animals have reduced lesion size and reduced parasite burdens. They also queried publicly available single cell datasets to compare to their murine single cell dataset and identified corresponding populations in human skin. I have no specific recommendations for the manuscript to be published in its current form; however I do think follow-up functional studies further characterizing the role of this macrophage population in liver and lung would be important given the visceral and mucosal forms of *L. major* infection and the fact that the authors found reporter+ TRM in these tissues.

Reviewer #2 (Remarks to the Author):

This paper by Lee et al investigates the mechanisms by which dermal tissue resident macrophages (TRMs) maintain their M2 like phenotype and promote *L. major* persistence. Authors report that IL-5 derived from ILC2 is required for maintaining dermal TRMs which are the source of TSLP required to maintain ILC2 as well as CCL24 required for eosinophil recruitment. Overall, authors report interesting findings of potentially high significance for the field but some data is not clear. Especially the careful analysis UMAP data included in the supplementary figure 1 show that TRMs are not the only source of TSLP. Following concerns need to be addressed to improve the scope of this work

Comments

More definitive proof that IL-5 derived from ILC2 will be obtained by adoptive transfer studies involving transfer of wild type ILC2 into *IL-5^{-/-}* mice to show that IL-5 produced by these cells is indeed required to maintain TRMs.

Fig 2B. Only one log reduction in parasite burden 12 weeks post infection and measurable lesions in *TSLPR^{-/-}* mice. This would suggest that lack of TSLP signaling has only modest impact on parasite loads and pathology suggesting other mechanisms be more important. This limitation needs to be discussed

Fig 3A. The UMAP is a combination of 4 different cell types. The supplementary figure that resolves the UMAP into each cell type shown in Supplementary figure 1 does not show TSLP. Hard to judge how much TSLP is produced in each cell type in presence or absence of infection.

Fig 3. 3A...3C only a subset of dermal TRMs produce TSLP. In the UMAP a subset of moDCs and endothelial cells also seem to be positive for TSLP.

Fig 3D shows that from the GEP knowledgebase MHCII neg macrophages produce the most TSLP, MHCII+ macrophages less so. In this paper that seems to be reversed. 3E shows non-hematopoietic cells also produce TSLP. Have you ruled out such cell types are not present in the dermal tissues.

Fig 4 E. Are the p values FDR corrected?

Fig 4E, In the heatmap of MHCII+ and MHCII- cells, the expression of TSLP is not indicated.

Fig5C. Why do the number of contacts between TRMs and ILC2s show a downward trend over the 80 minute observation period? Is this caused by diffusion effects?

Reviewer #3 (Remarks to the Author):

In the ms entitled “Dermis resident macrophages orchestrate localized ILC2-eosinophil circuitries to maintain their M2-like properties and promote non-healing cutaneous leishmaniasis” by Lee et al., the authors unravel that M2 dermal MF produce TSLP which promotes ILC2 recruitment which in turn induces eosinophil immigration via IL-4. The experimental setup is versatile and the data is very convincing.

I have only very few minor concerns:

ILC2 are the main skin-resident ILC and represent >70% of all NK/ILC under steady state conditions. However, the numbers are still low. Which percentage of ILC2 is absent in the absence of IL-4 producing ILC2 (Fig. 1c)?

What is the turnover of ILC2 and the DT injection frequency required to maintain an absence of these cells in Fig. 1C? The authors claim that the system may not allow for continuous depletion. Did they also monitor disease outcome in this setting?

In addition, the authors comment on the moDC being increased in this setting (Fig. 1C and again Fig. 7B), but do not comment or follow up on that. Fig. 1E: Did the authors also observe moDC increase as in the previous mouse model?

Fig. 2D: The frequency of IL-5+ ILC is very low already in controls and does not correlate to the numbers shown in Fig. 1B. Methodologically-dependent effect? Please explain.

Fig. 3F: The difference between mice with Il-4/13 deficiency in eosinophils with regard to Tslp expression is not really convincing. There seem to be two groups of expression levels in each genotype. Two different experiments?

All in all, in addition to the effect of type 2 cytokines (i.e. IL-4) having an important role on eosinophil recruitment, ILC2 may also promote M2-like MF and promote lesion resolution via wound healing-like mechanisms. This may be worth mentioning in the discussion.

REVIEWER COMMENTS

Reviewer #1 (Remarks to the Author):

In this manuscript, Lee et al report the role of TSLP and CCL24 producing, M2-like dermal tissue resident macrophages (TRM) in non-healing cutaneous lesions during *Leishmania major* infection in mice. This manuscript is very thorough and provides nice insights into macrophage heterogeneity and its importance in infectious immunity and inflammation. Of particular interest to the scientific community, the team generated *Ccl24-Cre* and *Tslp-flox* mice using CRISPR/Cas9 to specifically target this population, and showed that targeted animals have reduced lesion size and reduced parasite burdens. They also queried publicly available single cell datasets to compare to their murine single cell dataset and identified corresponding populations in human skin. I have no specific recommendations for the manuscript to be published in its current form; however I do think follow-up functional studies further characterizing the role of this macrophage population in liver and lung would be important given the visceral and mucosal forms of *L. major* infection and the fact that the authors found reporter+ TRM in these tissues.

- We appreciate the thoughtful comments. As mentioned, the *Ccl24-cre* mice are currently undergoing follow-up studies encompassing visceral and mucosal forms of *Leishmania* infection, as well as other types of immunopathology, including tumors and gut inflammation.

Reviewer #2 (Remarks to the Author):

This paper by Lee et al investigates the mechanisms by which dermal tissue resident macrophages (TRMs) maintain their M2 like phenotype and promote *L. major* persistence. Authors report that IL-5 derived from ILC2 is required for maintaining dermal TRMs which are the source of TSLP required to maintain ILC2 as well as CCL24 required for eosinophil recruitment. Overall, authors report interesting findings of potentially high significance for the field but some data is not clear. Especially the careful analysis UMAP data included in the supplementary figure 1 show that TRMs are not the only source of TSLP. Following concerns need to be addressed to improve the scope of this work

- We appreciate the overall positive comments. Regarding UMAP data in the supplementary figure 1, there was no TSLP annotation in this figure so we are confused about what the reviewer is referring to. We have included this analysis in the revised submission (Sup Fig. 1C), which reinforces the point that only the TRMs are source of TSLP in this model.

Comments

More definitive proof that IL-5 derived from ILC2 will be obtained by adoptive transfer studies involving transfer of wild type ILC2 into *IL-5^{-/-}* mice to show that IL-5 produced by these cells is indeed required to maintain TRMs.

- The suggested experiment was carried out and is shown in Figure 1F. The results illustrate that the adoptively transferred, expanded ILC2 from WT mice effectively restored the diminished numbers of TRMs and eosinophils, while reducing the elevated number of moDCs in infected *R5:Gata3^{fl/fl}* animals to levels comparable to those observed in WT controls. It is worth noting that we utilized *R5:Gata3^{fl/fl}* animals, which lack IL-5 producing ILC2, instead of *IL5^{-/-}* mice. This

choice was made to preemptively create the niche in the dermis for the adoptively transferred ILC2.

Fig 2B. Only one log reduction in parasite burden 12 weeks post infection and measurable lesions in TSLPR^{-/-} mice. This would suggest that lack of TSLP signaling has only modest impact on parasite loads and pathology suggesting other mechanisms be more important. This limitation needs to be discussed

- The following paragraph has been included in the discussion to address the specific limitation raised:

“Nonetheless, the 10-fold reduction in parasite burdens observed in the *Tslp*^{-/-} and *Ccl24-cre : Tslp*^{fl/fl} mice in comparison to the WT controls, did not approach the nearly 100-fold reduction observed in IL-5⁺ ILC2 deficient *R5 : Gata3*^{fl/fl} mice. This suggests a role for additional alarmin(s), possibly IL-18, that may activate a major subset of skin ILC2s, known as IL18R1-expressing ILC2s
1”

Fig 3A. The UMAP is a combination of 4 different cell types. The supplementary figure that resolves the UMAP into each cell type shown in Supplementary figure 1 does not show TSLP. Hard to judge how much TSLP is produced in each cell type in presence or absence of infection.

- To clarify the point raised, we have included a UMAP visualization of TSLP in the four different conditions (Sup Fig. 1C). This visualization clearly demonstrates that TSLP production by dermal TRMs remains consistent across all conditions, even though the relative frequency of TRMs in the UMAPs may vary due to immune cell infiltration following infection.

Fig 3. 3A...3C only a subset of dermal TRMs produce TSLP. In the UMAP a subset of moDCs and endothelial cells also seem to be positive for TSLP.

- We showed that the MHCII⁻ subset of TRMs is the main producer of TSLP in Figure 4.
- We observed that a subset of moDCs and endothelial cells exhibited positive *Tslp* expression, albeit at significantly lower levels compared to the dermal TRMs. To show that dermal TRMs were a critical source of TSLP, we conducted experiments using *Ccl24-Cre : Tslp*^{fl/fl} mice, which possess a selective TSLP deficiency in dermal TRMs. The infection outcome in these mice in terms of parasite burdens and pathology were similar to those observed in *Tslpr*^{-/-} mice.

Fig 3D shows that from the GEP knowledgebase MHCII⁻ macrophages produce the most TSLP, MHCII⁺ macrophages less so. In this paper that seems to be reversed. 3E shows non-hematopoietic cells also produce TSLP. Have you ruled out such cell types are not present in the dermal tissues.

- We are confused by this comment. In Figure 4 C and D, we demonstrate that MHCII-negative dermal TRMs produce far more TSLP in comparison to their MHCII-high counterparts.
- It is well-established that TSLP is typically produced by non-hematopoietic cells in various tissues in response to external stimuli, as illustrated in Figure 3E showing the comparison of *Tslp* transcription from various immune cells published by Immgen². By contrast we have observed that *Leishmania* infection in the dermis does not seem to induce a significant amount of TSLP

from non-hematopoietic cells, as identified through single-cell RNA sequencing. Instead, it primarily originates from dermal TRMs.

Fig 4 E. Are the p values FDR corrected?

- Yes, these values were corrected using the Bonferroni method to calculate the family-wise error rate.

Fig 4E, In the heatmap of MHCII+ and MHCII- cells, the expression of TSLP is not indicated.

- The average log2 fold change for TSLP was initially 0.43, but it was excluded from the differentially expressed gene (DEG) list in Figure 4E as it did not meet the threshold of avg log2FC > 0.5. Subsequently, we adjusted this threshold to avg log2FC > 0.4 to include TSLP in our DEG list in Figure 4E.

Fig5C. Why do the number of contacts between TRMs and ILC2s show a downward trend over the 80 minute observation period? Is this caused by diffusion effects?

- The 80-minute duration of intravital imaging led to photobleaching of Cy5-Manocept™, the fluorescent label for MR^{hi} dermal TRMs. This photobleaching contributed to the observed downward trend in the number of contacts between TRMs and ILC2.

Reviewer #3 (Remarks to the Author):

In the ms entitled “Dermis resident macrophages orchestrate localized ILC2-eosinophil circuitries to maintain their M2-like properties and promote non-healing cutaneous leishmaniasis” by Lee et al., the authors unravel that M2 dermal MF produce TSLP which promotes ILC2 recruitment which in turn induces eosinophil immigration via IL-4. The experimental setup is versatile and the data is very convincing.

- We appreciate the positive comments.

I have only very few minor concerns:

ILC2 are the main skin-resident ILC and represent >70% of all NK/ILC under steady state conditions. However, the numbers are still low. Which percentage of ILC2 is absent in the absence of IL-4 producing ILC2 (Fig. 1c)?

- In Figure 1C, we employed R5:ROSA26iDTR mice, which lack IL-5 producing ILC2, not IL-4 as mentioned by the reviewer.
- In Figure 1C, we can observe a 30% reduction in the number of ILC2s after DT-mediated depletion of IL-5-producing ILC2s. This implies that approximately 70% of ILC2s were not actively producing IL-5 during the infection. It is important to note that DT-mediated depletion using ROSA-DTR mice does not appear to be 100% efficient (Fig. 1B), and the use of R5 (IL5-Cre/TdTomato) transgenic mice could potentially result in under- or over-representation of actual IL-5 transcription levels.

What is the turnover of ILC2 and the DT injection frequency required to maintain an absence of these

cells in Fig. 1C? The authors claim that the system may not allow for continuous depletion. Did they also monitor disease outcome in this setting?

- We administered 20ng of DT intradermally on day 0 to deplete skin-resident ILC2s, and 200ng of DT intraperitoneally on day 2 and 4 post-infection to deplete ILC2s from circulation. This DT treatment effectively maintained ILC2 depletion up to day 12, implying that complete or significant turnover of ILC2s did not occur within this timeframe.
- It is reported that repeated treatments with DT can lead to the development of anti-DT neutralizing antibodies within two weeks, which will significantly reduce the efficacy of DT. Given that our infection model spans a period of at least 15-20 weeks, we did not conduct any further experiments to address the chronic infection outcomes in these mice.

In addition, the authors comment on the moDC being increased in this setting (Fig. 1C and again Fig. 7B), but do not comment or follow up on that. Fig. 1E: Did the authors also observe moDC increase as in the previous mouse model?

- The following paragraph has been added to the discussion to address this comment:

“We also observed an increase in the number of inflammatory monocytes/moDCs in *R5 : ROSA26iDTR* or *R5 : Gata^{1ff}* mice, as well as in *Ccl24-cre : Tslp^{1ff}* mice, which aligns with our previous findings in *eoCre : IL4/13^{1ff}* mice³. This suggests that TSLP/CCL24 originating from dermal TRMs has direct or indirect effects on monocyte recruitment, perhaps by creating an open resident macrophage niche. We have previously reported elevated levels of CCL3/4 (macrophage inflammatory protein 1 α , MIP1 α , and MIP1 β) in dermal TRMs from *eoCre : IL4/13^{1ff}* mice, which are known chemoattractants for monocytes³. Recruited monocytes and monocyte-derived cells likely contribute to the control of *L. major* infection, as in contrast to the dermal TRMs, these cells show up-regulated expression of inducible nitric oxide synthase^{4,5}.”

- The previously omitted data regarding monocyte infiltration has been incorporated into Figure 1E.

Fig. 2D: The frequency of IL-5+ ILC is very low already in controls and does not correlate to the numbers shown in Fig. 1B. Methodologically-dependent effect? Please explain.

- Figure 1B represents the percentage of tdTomato+ cells in *R5: ROSA26-LSL-tdTomato* mice, genetically labeled cells to express IL-5 at any point during their lifespan whether or not the protein is still present. By contrast, Figure 2D shows intracellular cytokine staining, which specifically identifies active IL-5 producers. The figure legend now clarifies this point.

Fig. 3F: The difference between mice with Il-4/13 deficiency in eosinophils with regard to Tslp expression is not really convincing. There seem to be two groups of expression levels in each genotype. Two different experiments?

- We agree that there appears to be a bimodality in gene expression levels for *Tslp* within dermal TRMs after infection. It is possible that the TRMs that are more distal to the infection site, including cells on the non-infected side of the ear, will show distinct transcriptional profiles. We anticipate that employing a spatial transcriptomic approach will provide some answers to this interesting question.

All in all, in addition to the effect of type 2 cytokines (i.e. IL-4) having an important role on eosinophil recruitment, ILC2 may also promote M2-like MF and promote lesion resolution via wound healing-like mechanisms. This may be worth mentioning in the discussion.

- The following paragraph has been added to the discussion in response to reviewer's suggestion.

“Given the reported interplay between ILC2s and alternatively activated macrophages in various contexts such as parasite infections, metabolic regulation, and tissue repair⁶, IL-5⁺ ILC2s may exert a direct influence on the number and M2-like phenotype of dermal TRMs by secreting type 2 cytokines. Our observations also indicate that antibody-mediated neutralization of TSLP leads to a reduction in the number of ILC2s and their interactions with dermal TRMs in infected skin. Since both cell types are localized within the perivascular niche in the dermis, it is possible that their close proximity and cross-regulation is facilitated by the direct release of TSLP by dermal TRMs. Interestingly, the co-localization of ILC2s with adventitial stromal cells (ASCs) around lung bronchi and larger vessels has been previously attributed to the production of IL-33 and TSLP by ASCs, which supports the accumulation and function of ILC2s in those settings⁷.”

1. Ricardo-Gonzalez, R.R. *et al.* Tissue signals imprint ILC2 identity with anticipatory function. *Nat Immunol* **19**, 1093-1099 (2018).
2. Gautier, E.L. *et al.* Gene-expression profiles and transcriptional regulatory pathways that underlie the identity and diversity of mouse tissue macrophages. *Nat Immunol* **13**, 1118-1128 (2012).
3. Lee, S.H. *et al.* M2-like, dermal macrophages are maintained via IL-4/CCL24-mediated cooperative interaction with eosinophils in cutaneous leishmaniasis. *Sci Immunol* **5** (2020).
4. Lee, S.H. *et al.* Mannose receptor high, M2 dermal macrophages mediate nonhealing *Leishmania major* infection in a Th1 immune environment. *J Exp Med* **215**, 357-375 (2018).
5. Stenger, S., Thuring, H., Rollinghoff, M. & Bogdan, C. Tissue expression of inducible nitric oxide synthase is closely associated with resistance to *Leishmania major*. *J Exp Med* **180**, 783-793 (1994).
6. von Moltke, J. & Locksley, R.M. I-L-C-2 it: type 2 immunity and group 2 innate lymphoid cells in homeostasis. *Curr Opin Immunol* **31**, 58-65 (2014).
7. Dahlgren, M.W. *et al.* Adventitial Stromal Cells Define Group 2 Innate Lymphoid Cell Tissue Niches. *Immunity* **50**, 707-722 e706 (2019).

REVIEWERS' COMMENTS

Reviewer #2 (Remarks to the Author):

Authors have addressed my concerns in the revised manuscript. Additional experiments strengthen the conclusions.

Reviewer #3 (Remarks to the Author):

None